# Parallels and contrasts between the cnidarian and bilaterian maternal-to-zygotic transition are revealed in *Hydractinia* embryos

Taylor N. Ayers[1], Matthew L. Nicotra[2], Miler T. Lee[1]*

1 Department of Biological Sciences, University of Pittsburgh, Pittsburgh Pennsylvania, United States of America, 2 Thomas E. Starzl Transplantation Institute, University of Pittsburgh, Pittsburgh, Pennsylvania, United States of America

* miler@pitt.edu

**Data Availability Statement:** Sequencing data are available in the Gene Expression Omnibus (GEO) under accession number GSE232065. Additional data files including the de novo assembled

## Abstract

Embryogenesis requires coordinated gene regulatory activities early on that establish the trajectory of subsequent development, during a period called the maternal-to-zygotic transition (MZT). The MZT comprises transcriptional activation of the embryonic genome and post-transcriptional regulation of egg-inherited maternal mRNA. Investigation into the MZT in animals has focused almost exclusively on bilaterians, which include all classical models such as flies, worms, sea urchin, and vertebrates, thus limiting our capacity to understand the gene regulatory paradigms uniting the MZT across all animals. Here, we elucidate the MZT of a non-bilaterian, the cnidarian *Hydractinia symbiolongicarpus*. Using parallel poly (A)-selected and non poly(A)-dependent RNA-seq approaches, we find that the Hydractinia MZT is composed of regulatory activities similar to many bilaterians, including cytoplasmic readenylation of maternally contributed mRNA, delayed genome activation, and separate phases of maternal mRNA deadenylation and degradation that likely depend on both maternally and zygotically encoded clearance factors, including microRNAs. But we also observe massive upregulation of histone genes and an expanded repertoire of predicted H4K20 methyltransferases, aspects thus far particular to the Hydractinia MZT and potentially underlying a novel mode of early embryonic chromatin regulation. Thus, similar regulatory strategies with taxon-specific elaboration underlie the MZT in both bilaterian and non-bilaterian embryos, providing insight into how an essential developmental transition may have arisen in ancestral animals.

## Author summary

Eggs are preloaded with genetic instructions encoded as mRNA, which not only drive the first stages of embryogenesis, but also induce the production of new mRNA from the embryonic genome that supplant the egg-inherited ones. Many different species undergo this process, but the extent to which they share common mechanisms to regulate the transition from egg to embryonic mRNA is still unclear, especially among very distant animal relatives to humans. We have investigated this transition in Hydractinia, a representative

transcriptome are available at OSF, https://osf.io/jpq3v/.

**Funding:** M.T.L was supported by NIH grant R35GM137973. M.L.N. was supported by NSF grants 1557339 and 1923259. The funders had no role in study design, data collection and analysis, decision to publish, or preparation of the manuscript.

**Competing interests:** The authors have declared that no competing interests exist.

of the Cnidaria phylum that also includes jellyfish and sea anemones. Using optimized methods to measure mRNA quantities during early Hydractinia development, we find an overall trajectory of mRNA change closely resembling what occurs in embryos of more commonly studied animals, as well as evidence for similar mechanisms regulating these changes. However, we also uncovered surprising patterns of gene expression for chromosome structural genes (histones) and enzymes that chemically modify them, suggesting that Hydractinia embryos regulate their genomes in a novel way compared to what has previously been observed. Our findings provide insight into how our common animal ancestor may have regulated early development, but also reveal a new aspect of gene regulation that warrants further investigation in other contexts.

## Introduction

Across sexually reproducing organisms, the earliest stages of embryonic development are guided by cellular components inherited from the egg, including a large maternal RNA contribution. Eventually, new RNA are transcribed from the embryonic genome that will supplant the maternal RNA and assume developmental control as the embryo proceeds through stem cell induction and gastrulation. This maternal-to-zygotic transition (MZT) comprises two activities: the activation of the embryonic genome (zygotic genome activation, ZGA) and the removal of the maternal RNA contribution (maternal clearance), which combine to reprogram the embryonic transcriptome away from an egg identity to an embryonic stem cell identity [1]. The regulatory logic and molecular bases for these processes remain to be fully deciphered.

Inquiry into the animal MZT has focused primarily on one side of the animal phylogeny, the bilaterians. Investigations in classical model bilaterian embryos, including sea urchin, Drosophila, *C. elegans*, Xenopus, zebrafish, and mouse has revealed wide divergence in the timing and mechanisms underlying events of the MZT, though some common themes have emerged (reviewed in [1,2]). In many of these animals, genome activation occurs only after a period of transcriptional quiescence following fertilization [3,4]. For faster-dividing taxa such as Drosophila, Xenopus, and zebrafish, an initially unfavorable concentration of transcriptional repressors relative to the quantity of DNA (the nucleocytoplasmic ratio) must first be overcome by undergoing several rounds of genome replication [4–8]. This effect is at least partly achieved by maternally derived histone proteins, which have been found to contribute to genome activation repression [9–13]. Genome activation is gradual in these taxa, starting with activation of a smaller subset of genes during cleavage and early blastula stages, then growing to encompass up to one-third of the genes in the genome by early gastrula stages [1].

Maternal activators can also need time to gain competence. A large portion of the maternal contribution is stored in the oocyte with short poly(A) tails, and cytoplasmic readenylation of these mRNA is triggered only after fertilization or egg activation [14–16], which then facilitates increased translation of these messages [16–19]. Key maternal transcription factor mRNA are among the transcripts experiencing readenylation, and post-transcriptional regulation of these along with chromatin modifying factors also contributes to the timing of genome activation and embryonic transcriptome reprogramming [17,18].

Many maternal mRNA will also undergo regulated clearance through translation repression, decapping, deadenylation, and nucleolytic degradation [20,21]. Intrinsic features are correlated with differential maternal mRNA stability, including the frequency of rare codons [22,23] and the presence of the covalent RNA base modifications N6-methyladenosine ($m^6A$) and 5-methylcytosine ($m^5C$) [24–26]. RNA-binding proteins (RBPs) play major roles in

mediating clearance, such as Smaug, Pumilio and BRAT in Drosophila [27–30], and AU-rich element binding proteins across multiple species [31–34], recognizing target mRNA via sequence and structural elements encoded in their 3' untranslated regions (3' UTRs). micro-RNAs have also emerged as maternal clearance regulators in zebrafish (miR-430), Xenopus (miR-427), and Drosophila (miR-309) [35–38]. These and other factors recruit deadenylation machinery such as CCR4-NOT [39] and PARN [40].

The extent to which these regulatory themes and mechanisms are shared with non-bilaterian animals is still largely unknown. In the ctenophore *Mnemiopsis leidyi*, genome activation and maternal clearance were inferred to occur within three cell cycles after fertilization, according to an RNA-seq timecourse [41]. Among cnidarians, transcriptomic profiling has detected hundreds of genes up- and down-expressed in embryonic stages spanning gastrulation in the coral species *Acropora digitifera*, *Favia lizardensis*, *Ctenactis echinata* [42–44], *Acropora millepora* [45] and *Montipora capitata* [46]. In the starlet sea anemone *Nematostella vectensis*, a major wave of gene activation occurs between the early and mid-blastula stages, with potentially some genes activating earlier, while 15% of the maternal contribution declines [47], implying that both genome activation and maternal clearance are occurring at similar early embryonic stages to many egg-laying bilaterians.

A major shortcoming of these previous studies is the use of poly(A)+ transcriptomic methods, which cannot distinguish changes in poly(A) tail length from changes in mRNA levels and underestimate the expression of non-adenylated transcripts [2,48,49]; thus, the magnitude and timing of the MZT in any cnidarian remain to be clarified. As Cnidaria is the sister phylum to Bilateria, it occupies a valuable taxonomic position to infer regulatory features of an ancestral animal from which derived mechanisms in extant taxa may have originated [50].

Here, we precisely measure the timing and extent of the MZT in the colonial hydrozoan *Hydractinia symbiolongicarpus*. Hydractinia is a cnidarian model for the study of regeneration, immunology, and more recently embryonic development among other topics [51]. Unlike many other cnidarians, Hydractinia can be induced to spawn regularly to produce large numbers of embryos amenable to experimentation. After fertilization, symmetric cleavages form a transient 16-cell blastula at ~2 hours post fertilization (h.p.f.) prior to the onset of gastrulation [52]. Genome activation has been estimated to occur by the 64-cell stage by nuclear staining for 5-Ethynyl Uridine (EU) incorporation into nascent RNA [53]. The *H. symbiolongicarpus* transcriptome has previously been profiled in adult tissues [54,55] and germ cells [56] and a recent genome assembly available on the Hydractinia Genome Project Portal (https://research.nhgri.nih.gov/hydractinia/) has begun to accelerate further genomic inquiry.

We have generated an RNA-seq timecourse of *H. symbiolongicarpus* embryonic development, using both poly(A) selection and a custom-made ribosomal RNA (rRNA) depletion strategy to measure RNA levels independent of poly(A) tail length. We find that a large wave of maternal mRNA polyadenylation precedes genome activation in the early embryo. Transcription of histone genes overwhelmingly dominates the first phase of genome activation, leading to an 18-fold increase in histone mRNA concentration by 4 h.p.f. in the early gastrula, a magnitude not previously observed in other taxa. Transcriptional activation is also detected for thousands of genes that are already maternally provided, in addition to transposon-related sequences and 22 different predicted histone H4 lysine 20 (H4K20) methyltransferases, which suggests that dynamic H4K20 methylation may contribute to embryonic chromatin reprogramming during the MZT. Maternal mRNA clearance by deadenylation is detected in waves starting at 2 h.p.f., but degradation is delayed until after genome activation. Finally, we provide computational evidence that de novo transcribed miRNAs may contribute to the clearance of a small number of maternal mRNA. Our findings reveal strong parallels between the cnidarian

and model bilaterian MZT, suggesting that shared gene regulatory paradigms underlie early embryogenesis programs across the animal tree.

## Results

### The Hydractinia maternal mRNA contribution undergoes extensive poly (A) tail length changes, prior to genome activation

To determine the timing of genome activation in Hydractinia, we needed to ensure that we could distinguish changes in absolute mRNA levels from changes in poly(A) tail length in the early embryo. To this end, we designed custom *H. symbiolongicarpus* antisense oligomers for rRNA-depletion RNA-seq, with the aid of our Oligo-ASST tool [49], targeting empirically determined nuclear and mitochondrial rRNA sequences (S1A–S1D Fig and S1, S2 Tables). We then performed RNA-seq on unfertilized eggs, fertilized eggs (0.5 h.p.f.), and embryos every hour from 1 to 7 h.p.f. at 23˚C, spanning the onset of gastrulation [52] (Fig 1A). Both rRNA-depleted and poly(A)+ libraries were built in parallel for each sample. As a reference for later development, we additionally constructed poly(A)+ libraries during larval stages, 24, 48, and 72 h.p.f.

We found expression evidence for ~18,000 Hsym v1.0 protein-coding genes at ≥1 transcript per million (TPM) in at least one timepoint. In the unfertilized egg, 11,853 genes are expressed ≥1 TPM (S3 Table), indicating that the maternal contribution spans ~54% of annotated genes, comparable to other animals [1].

In the rRNA-depleted libraries, the transcriptomes are nearly identical until 2 h.p.f. (8–16 cells), when we first see subtle activation of 20 genes (Fig 1B, top and S1, S4, S5 Tables) (here and henceforth, we use the terms "activate" and "activation" to mean transcriptional activation). This confirms that *H. symbiolongicarpus* embryos indeed undergo delayed genome activation, similar to most bilaterian animals [1]. In contrast, in the poly(A)+ libraries, 471 genes have significantly higher read counts at 2 h.p.f. compared to egg stage by DESeq2 (adjusted $P < 0.05$), and 1331 genes have significantly lower counts (Fig 1B, bottom). Taken together with the rRNA-depletion libraries, which were each built in parallel on the same biological samples as the poly(A)+ libraries, we conclude that changes in poly(A) tail length are driving most of the measured transcriptome changes by 2 h.p.f. This indicates that cytoplasmic readenylation of the maternal contribution, previously observed across bilaterian embryos, also occurs in a cnidarian embryo, along with what is likely deadenylation of another subset of mRNA as part of the onset of maternal clearance.

By 3 h.p.f. (32–64 cell early gastrula), >2500 genes have significant differences in the poly (A)+ libraries (1016 genes increasing, 1656 genes decreasing), but the rRNA-depleted libraries show only 151 genes whose levels have significantly increased, reflecting the progression of genome activation. And by 4 h.p.f., only 465 genes are significantly activated in the rRNA-depleted libraries, compared with 2152 genes up in the poly(A)+ libraries (Fig 1B, far right). Together, these data illustrate the scope of genome activation in Hydractinia, which begins in the blastula and ramps up as gastrulation proceeds, concurrent with a broader set of dynamic poly(A) tail length changes in the maternal contribution, similar to model bilaterians such as Drosophila and zebrafish.

### Thousands of re-activated maternal genes are revealed by intronic RNA-seq coverage

To detect subtler signals of embryonic genome activation, we quantified intronic sequencing read coverage. In other animals, a large proportion of de novo transcription at genome

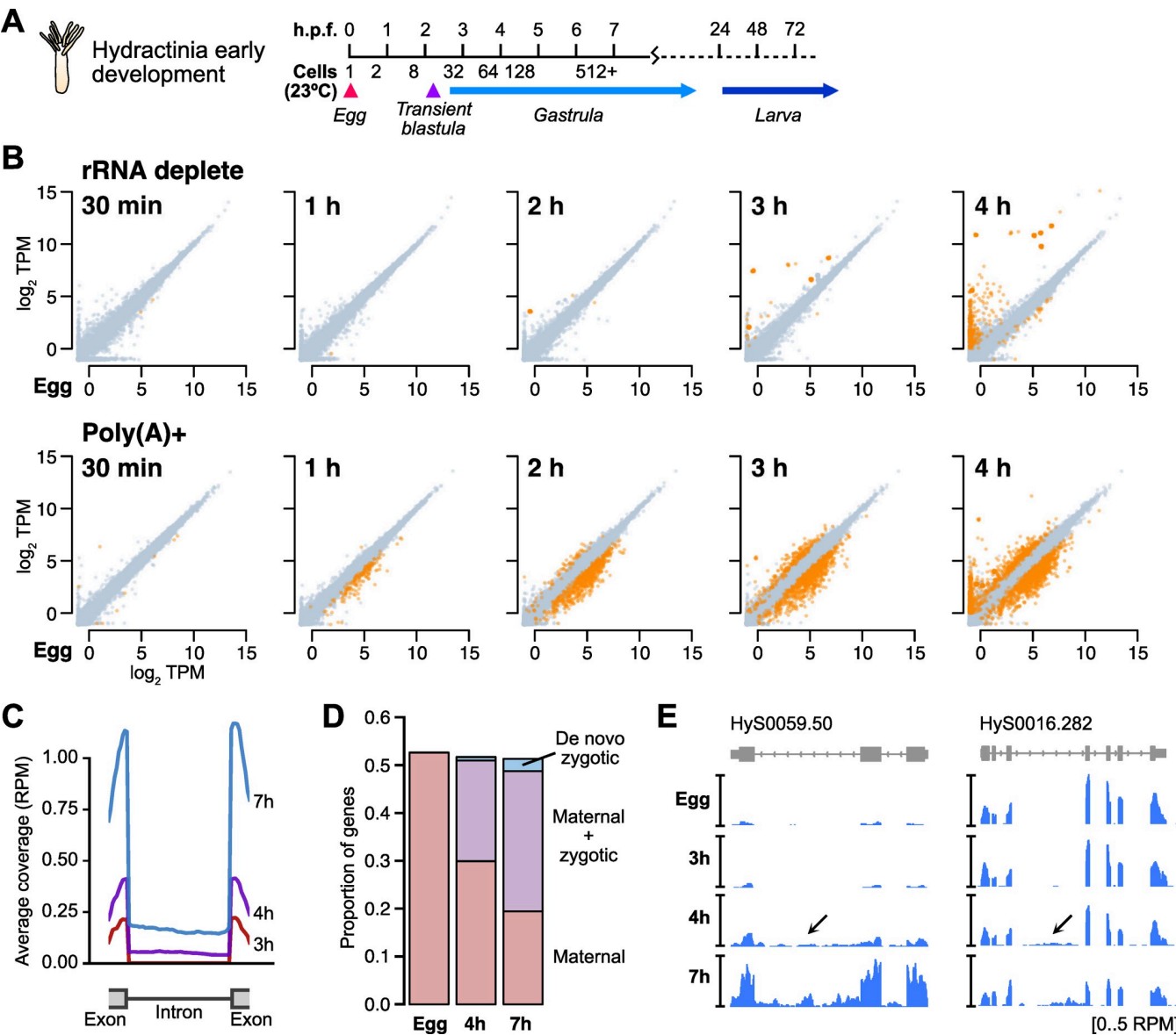

**Fig 1. RNA-seq with rRNA depletion reveals the timing and extent of genome activation.** (**A**) Schematic of Hydractinia embryonic development comparing hours post fertilization (h.p.f.) with approximate cell count, based on Kraus et al [52]. Gastrulation likely completes prior to 20 h.p.f. (**B**) Biplots of $\log_2$ TPM RNA-seq expression levels between unfertilized eggs (x axis) and subsequent developmental time points. Top panels represent RNA-seq with rRNA depletion, bottom panels represent RNA-seq with poly(A)+ selection. Genes >2-fold significantly different (DESeq2 adjusted P < 0.05, TPM > 1) are highlighted in orange. (**C**) Meta plots of rRNA-depleted RNA-seq coverage over introns of activated genes, as measured 3, 4, and 7 hours post fertilization. (**D**) Stacked barplots showing the proportion of annotated genes at different time points that are maternally provided with no additional activation (pink), maternally provided with additional expression from the embryonic genome (purple), and de novo transcribed in the embryo (blue). (**E**) Browser tracks showing rRNA-depleted RNA-seq intronic coverage appearing at 4 hours post fertilization (arrows). TPM = transcripts per million. RPM = reads per million.

activation occurs for genes that already have a maternal RNA contribution, which can be difficult to dissect apart in RNA-seq data [2,18,31,48,57–59]. Previously, we demonstrated that RNA-seq reads mapping to intronic regions can detect de novo pre-mRNA production in maternal genes where increases in exonic read coverage are too modest to robustly quantify [18,59].

We aligned our rRNA-depleted libraries to the genome, then quantified reads mapping to exons versus introns. Intronic counts were defined as reads mapping uniquely and

unambiguously within a gene's boundaries, minus reads aligning any exon. Among significantly activated Hsym genes (DESeq2 adjusted $P < 0.05$) with minimal maternal contribution ($< 0.5$ TPM), we observe a bulk increase in both exonic and intronic RNA-seq coverage spanning exon-intron junctions starting at 4 h.p.f. (Figs 1C and S1E), indicating that we are able to detect pre-mRNA signal.

With our increased sensitivity, we found significant transcriptional activation at 4 h.p.f. of 4,647 genes (DESeq2 adjusted $P < 0.05$) that already had a maternal contribution ($\geq 1$ TPM in the egg) (Fig 1D and S1 Table), of which 98% failed to be detected using only exonic RNA-seq signal. By 7 h. p.f., this increases to a total of 6,490 maternal genes that are significantly re-expressed (Fig 1D, 1E). These data demonstrate that although the majority of de novo transcription comprises a small subset of highly activated genes detectable by exon level increases, genome activation is nonetheless widespread in Hydractinia, consistent with what has been observed in other animals.

## Arrays of histone genes activate first

To infer the functions of the most strongly activated genes in the *H. symbiolongicarpus* genome, we performed protein BLAST alignments against the UniProtKB/Swiss-Prot database to identify potential homologs. Nearly all of the earliest-activated genes appear to encode histone proteins: 18 of the 20 genes detected at 2 h.p.f. correspond to copies of the histone H4 gene, accompanied by a linker histone H1 gene and a repetitive sequence that may be misannotated as a protein-coding gene (S4 Table). At 3 h.p.f., this expands to include 123 copies of histones H2A, H3, H4, and H1 linker, and 150 additional histone genes at 4 h.p.f. including H2B. Many of these histone genes are found clustered in large arrays across several different genomic scaffolds. The largest cluster is on scaffold HyS0385 (Fig 2A), consisting of 44 predicted histone genes, interspersed with likely RNA Polymerase III transcribed genes (U1, U2, and 5S rRNA), similar to what has been found in sister species *H. echinata*, where evidence suggests there are in fact hundreds of tandemly duplicated histone clusters in the genome [60].

Although there is a maternal contribution of mRNA encoding all four core histone proteins and H1 summing to 1.8% of the egg transcriptome, total histone mRNA levels double by 3 h.p. f. and are 18-fold increased by 4 h.p.f., expanding to take up one-third of the embryonic protein-coding transcriptome (Fig 2B, 2C). This increase is primarily driven by activation of the core replication-dependent histones H2A.1, H2B.1, H3.1, H4.1 and H1.1, whose identities were determined by sequence homology to *H. echinata* [60] (Fig 2B and S1, S3 Tables).

Upregulation of histone mRNA expression in the early embryo has been observed in other taxa [61,62], but to more comprehensively compare these trends, we quantified histone expression in published RNA-seq embryonic datasets also built using rRNA depletion [63–67], since replication-dependent histones lack poly(A) tails (Fig 2C, 2D and S6, S7 Tables). We find that *H. symbiolongicarpus* is notable for having such a low starting concentration of maternal histone mRNA, followed by a massive increase at the onset of genome activation. In zebrafish, Xenopus, and Drosophila, maternal histone mRNA levels are higher and increases seem to occur only after the major wave of genome activation has begun, whereas in *H. symbiolongicarpus* histone mRNA transcription seems to precede strong activation of other genes. In contrast, the initial burst of transcription in zebrafish and Xenopus embryos is focused on a different highly multi-copy gene locus, the microRNA *mir-430* and *mir-427* clusters, respectively (Fig 2D). Sea urchin (*Strongylocentrotus purpuratus*) embryos also appear to transcribe histones first (Fig 2D), though they reportedly lack a transcriptionally quiescent phase prior to genome activation [1,68].

Prior to the transcriptome dominance by the core histone mRNA, some histone variants are relatively more prevalent in the maternal contribution, including H2A.Z, H2A.X, H3.3, and H1.2 (Fig 2B). Of note, we identified a predicted additional maternal H1 variant not

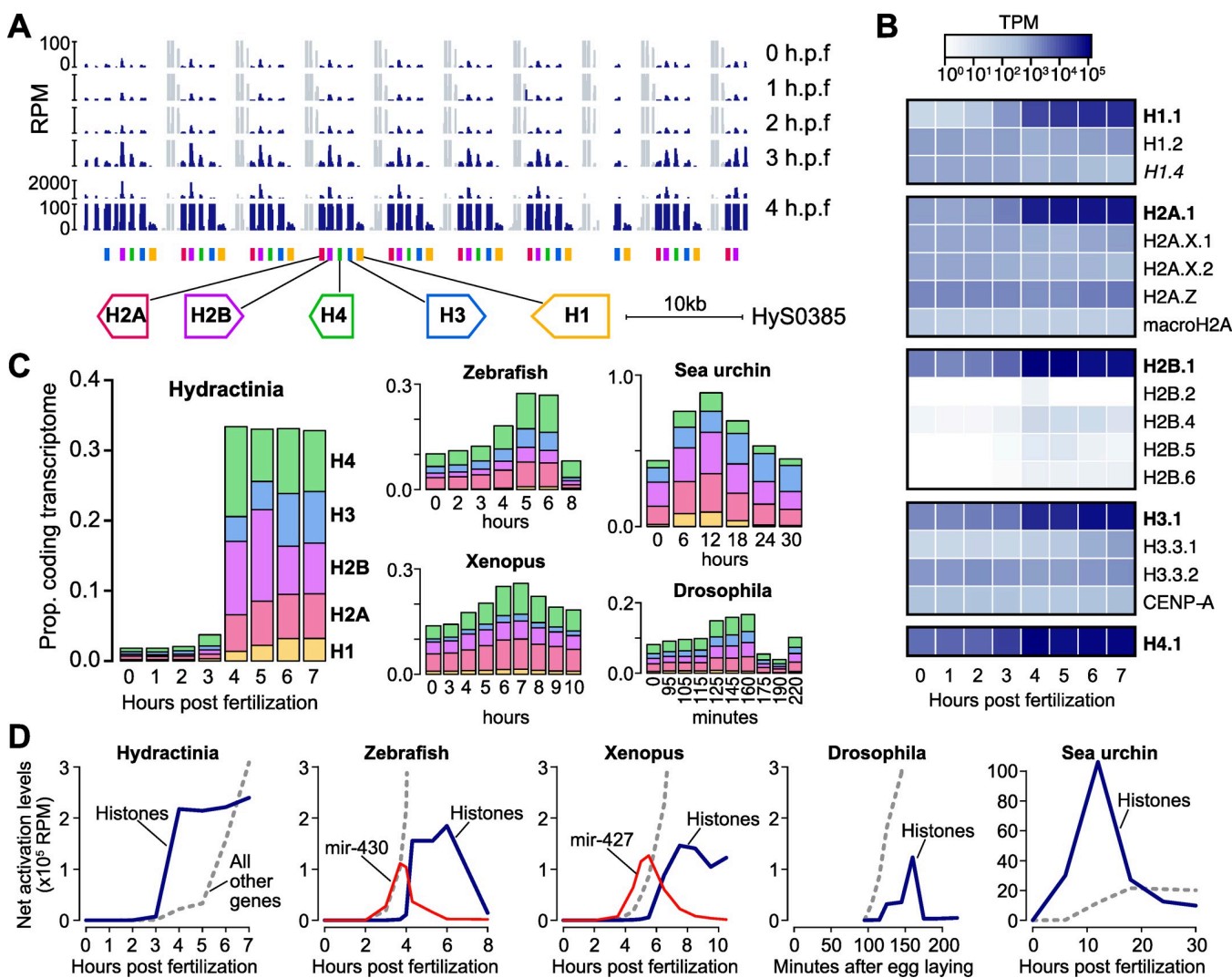

**Fig 2. Histone genes are strongly activated early.** (A) Browser tracks showing a cluster of tandemly repeated histone genes and the increase in RNA-seq coverage at 3–4 hours post fertilization (h.p.f.). The scale at 4 h.p.f. is discontinuous. Gray regions represent coverage of interspersed non-coding genes that are maternally provided. (B) Heatmap ($\log_{10}$ scale) showing strong expression increase of the core replication-dependent histones (bolded) over time. (C) Stacked barplots showing the proportion of the transcriptome that encodes each of the 5 histone classes (linker H1, H2A, H2B, H3, and H4) over time in embryos of Hydractinia (left) and four bilaterian species. (D) Net activation of all histones (blue lines) compared to all other genes (gray dashed lines) relative to maternal levels during early development in five species. Activation trajectories of microRNAs miR-430 and miR-427 are shown in red for zebrafish and Xenopus, respectively. TPM = transcripts per million, RPM = reads per million.

previously annotated in *H. echinata*, which we named H1.4 according to the *H. echinata* nomenclature [60] (*H. echinata* H1.3 is thought to be a pseudogene and does not have a match in the *H. symbiolongicarpus* Hsym v1.0 transcriptome). H1.4 has BLAST protein similarity to *Hydra vulgaris* H1A-like (XP_047140340.1, E = $2\times10^{-15}$) and in the current genome assembly is encoded as two identical 3-exon genes (HyS0030.125 and HyS2746.2). Its mRNA levels decline ~2-fold after genome activation (Figs 2B and S2).

These data demonstrate that the initial phase of transcriptome remodeling at the Hydractinia maternal-to-zygotic transition is driven by a major increase in histone mRNA concentrations in a pattern unlike what we observe in model bilaterians, which perhaps correlates with a unique mode of chromatin remodeling activity in the developing Hydractinia embryo.

## Twenty-two predicted H4K20 methyltransferase also activate early

Besides histone mRNA, the majority of other strongly activated genes by 4 h.p.f. seem to correspond to retrovirus-related open reading frames, e.g. HyS4714.1, HyS4716.1, and HyS4654.1 (S4 Table), suggesting early activation of transposable element expression at genome activation, as has been observed in other taxa [69]. Of the remaining genes activated to $\geq\sim$50 TPM, only a predicted ammonium transporter (HyS0101.18, 70 TPM), a forkhead-box transcription factor (HyS0062.47, 61 TPM), and a likely Sp5 transcription factor homolog (HyS0057.112, 47 TPM) had strong BLAST similarity to animal proteins (BLAST E-values of $2x10^{-85}$, $8x10^{-30}$, and $4x10^{-65}$ respectively against the UniProtKB/Swiss-Prot database) (S4 Table).

We additionally identified early de novo activation of 13 genes with BLAST similarity to the histone H4 lysine 20 (H4K20) methyltransferase *KMT5A/SETD8* (Fig 3A, 3B and S4 Table). Unlike the histone genes, these genes are not adjacent to each other in the genome assembly, and their amino acid sequences are highly divergent, though all of them share significant homology to the KMT5A/SETD8 SET domain by CD-Search (E-value = $1.9x10^{-10}$ to $1.3x10^{-39}$) (Fig 3B and S8 Table). Further investigation revealed as many as 65 annotated genes with significant BLAST similarity to KMT5A, including two with strong maternal contributions and nine additionally activated by 7 h.p.f. (Fig 3A and S3, S8 Tables). Only maternal HyS0034.48, at 315 amino acids in length with its SET domain on the C-terminal end, resembles the architecture of other species' KMT5A [70], whereas most of the other genes encode N-terminal SET domains, in addition to other functional domains in the longer genes (Fig 3B and S8 Table).

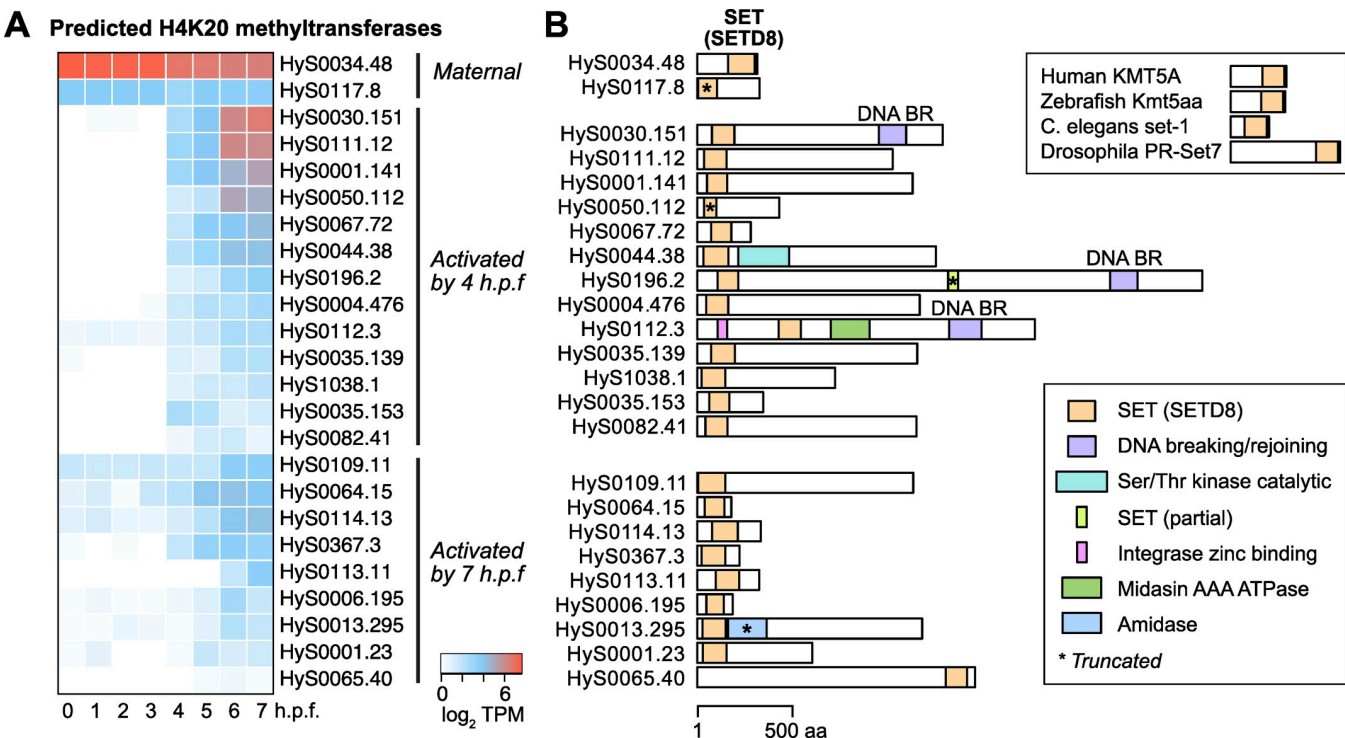

**Fig 3. Predicted H4K20 methyltransferase genes broadly activate. (A)** Heatmap showing expression levels of predicted H4K20 methyltransferase genes over time, grouped according to when activation is observed. **(B)** Protein domain schematics for the predicted H4K20 methyltransferases showing positions of the SETD8 SET domain (orange), as predicted by CD-Search. Other domains are indicated in other colors as shown in the legend. Reference H4K20 methyltransferases in other species are shown in the upper right. TPM = transcripts per million.

To investigate whether the expansion of predicted H4K20 methyltransferase genes is specific to Hydractinia, we queried the transcriptomes of the cnidarians Hydra (also a hydrozoan along with Hydractinia) and the more distantly related Nematostella. We found that Hydra also has an expanded repertoire of 24 genes with significant similarity to the SETD8 SET domain by CD-Search (E-value = $3.2x10^{-4}$ to $1.2x10^{-76}$) (S8 Table). Six have evidence for orthology to Hydractinia genes by reciprocal best BLAST alignment, but of those all but one has a truncated SET domain, suggesting functional divergence (S3A Fig). In contrast, we found only a single KMT5A gene in Nematostella (S3A Fig), indicating that expansion of H4K20 methyltransferases is not a shared feature of all cnidarians. We additionally looked in the ctenophore *M. leidyi*, which revealed two gene models in the reference transcriptome with C-terminal SETD8 SET domains, along with seven additional hits among the Mnemiopsis Genome Project unfiltered protein models, though all but one have 100% sequence identity to the first two genes and thus may represent alternative isoforms or misannotations (S3B Fig and S8 Table).

Among bilaterians, we did find an unexpectedly large number of KMT5A-like genes in the sea urchin *S. purpuratus*– 110 RefSeq models spanning 61 genes (S8 Table). Twenty-four are expressed during early *S. purpuratus* embryogenesis, in a pattern resembling what we observe in Hydractinia: two genes with canonical C-terminal SETD8 SET domains are strongly maternally provided but their expression declines over development, while the remaining genes mostly have N-terminal SET domains and are activated in later stages (S4A, S4B Fig and S6 Table). Furthermore, we found strong sequence similarity between the Hydractinia and *S. purpuratus* longer genes, encompassing both the N-terminal SET domains (BLAST E-value $< 3x10^{-10}$) and the C-terminal ends (BLAST E-value $< 3x10^{-43}$) (S4C Fig), suggesting deep homology of these novel predicted KMT5A genes.

KMT5A/SETD8 catalyzes monomethylation of H4K20 (H4K20me1) [71] and is associated with DNA replication and differential chromatin accessibility, though its roles in gene expression are not well understood [70]. Recent evidence suggests that H4K20me1 is associated with increased chromatin accessibility at gene bodies, but is also readily converted to repressive trimethylated H4K20 [72]. Gungi et al showed that H4K20 monomethylation via SETD8 is involved in both locus-specific repression and transcriptional activation during regeneration and axis patterning in Hydra [73]. Together with our findings, this suggests that H4K20 methylation may be a common feature of chromatin regulation among hydrozoan stem cells to achieve specific gene activation, which may also be selectively conserved in other animals such as sea urchin.

## Repetitive and transposon-related RNA are pervasive during genome activation

To further investigate the breadth of genome activation, we first profiled RNA-seq coverage in 10kb-windows to identify any unannotated genomic regions with evidence for strong activation. All windows with average increases of >3-fold are attributable to histone genes, confirming that coordinated histone gene transcription accounts for the bulk of early genome activation (S5A Fig and S1 Table).

Next, to identify activation of previously unannotated genes, we performed RNA-seq guided transcriptome assembly using StringTie [74] to augment the Hsym v1.0 gene models. This yielded 32,766 transcript models for 25,808 genes, of which 13,179 (40%) were existing or augmented Hsym v1.0 models, 6203 (19%) were predicted alternative isoforms of Hsym v1.0 models, and the rest (41%) novel (S9 Table). We annotated the novel gene models against several non-coding and coding databases (see Methods), and found that 4997 (37%) encoded

known non-coding RNA classes, including small nuclear RNA, tRNA, and 5S rRNA, which we had not targeted with rRNA depletion oligos. Levels for these in our rRNA-depleted samples appear consistent between stages (S5B Fig and S1 Table).

We quantified expression of the remaining gene models and found largely the same trends as before (S4 Table). Among the de novo Stringtie-annotated genes significantly activated by 4 h.p.f. to $\geq 5$ TPM (DESeq2 adjusted $p < 0.05$), 18 are likely histone genes by BLAST similarity, but the majority are repetitive or transposon-related sequences (S4 Table). The remaining predicted genes largely have poor coding potential and no significant BLAST similarity to UniprotKB/Swiss-Prot proteins, suggesting that they may encode novel non-coding RNAs (S4, S9 Tables). Although assigning RNA-seq reads specifically to repetitive genomic loci can be a challenge, as a general class, we find that repetitive sequences, including the highly multi-copy histone genes, feature prominently in the initial wave of genome activation in Hydractinia.

The prevalence of repetitive and transposon-related sequences in Hydractinia genome activation is reminiscent of mammalian genome activation, during which elevated transcription of transposable elements has been observed [69]. A family of murine endogenous retroviruses (MERVL) is activated by the pioneer transcription factor Dux in mouse 2-cell embryos, and MERVL sequences have likely been co-opted as part of the mouse genome activation network [75,76]. A similar phenomenon occurs in human embryos involving a different endogenous retrovirus family, which suggests the convergent evolution of retrotransposon-regulated genome activation in these taxa [76,77]. This raises the possibility that regulatory sequences driving Hydractinia genome activation may have transposable-element origins as well.

## Patterns of stage-specific gene expression emerge across the timecourse

We observe subsequent phases of stage-specific expression when we compare our early time-points to the later larval timepoints. ~3000 genes exhibit non-ubiquitous temporal expression, and many of the ~750 genes that peak in the blastula/gastrula are no longer strongly expressed at 24 h.p.f. (Fig 4A and S10 Table). Conversely, 1212 genes presumably associated with later development peak in our dataset at 24 h.p.f., though their expression may have begun during intervening time points for which we lack data.

To identify functional enrichment among these gene groups, we inferred Gene Ontology (GO) terms for each *H. symbiolongicarpus* gene from its best SwissProt BLAST hit (E-value $< 1 \times 10^{-5}$). We measured differential occurrence of each term among genes peaking in egg, gastrula (4–7 h.p.f.), 24, 48, or 72 h.p.f. using Chi-squared tests of independence and found 132 terms with significant stage-specific occurrence (FDR adjusted $P < 0.05$, 4 d.o.f.). Among the top terms enriched for the egg were functions related to DNA replication and repair and Wnt signaling; chromatin regulation in the gastrula, driven exclusively by histone genes; ERK1/2 MAP kinase signaling, ion transport, and collagen biosynthesis at 24 h.p.f.; protein metabolism at 48 h.p.f.; and neurotransmission at 72 h.p.f. (Fig 4A and S10 Table).

We also looked for stage-specific transcription factor expression. Focusing first on potential genome activators, we interrogated the maternal contribution for highly expressed genes whose predicted homologs are annotated with the GO terms "DNA-binding transcription factor activity" or "sequence-specific DNA binding." Among the genes with the most asymmetric expression compared to later stages (>5-fold higher in the maternal contribution), several classes of DNA binding domains are represented, including winged-helix, zinc finger, basic helix-loop-helix, and forkhead (Fig 4B and S10 Table). Notably, two predicted Sox homologs are maternally expressed with similarity to vertebrate Sox2 and Sox4 (BLAST E = $1.49 \times 10^{-44}$ and $1.21 \times 10^{-32}$, respectively). The previously characterized POU-domain transcription factor Polynem, which is associated with embryonic and adult stem cells in *Hydractinia echinata*

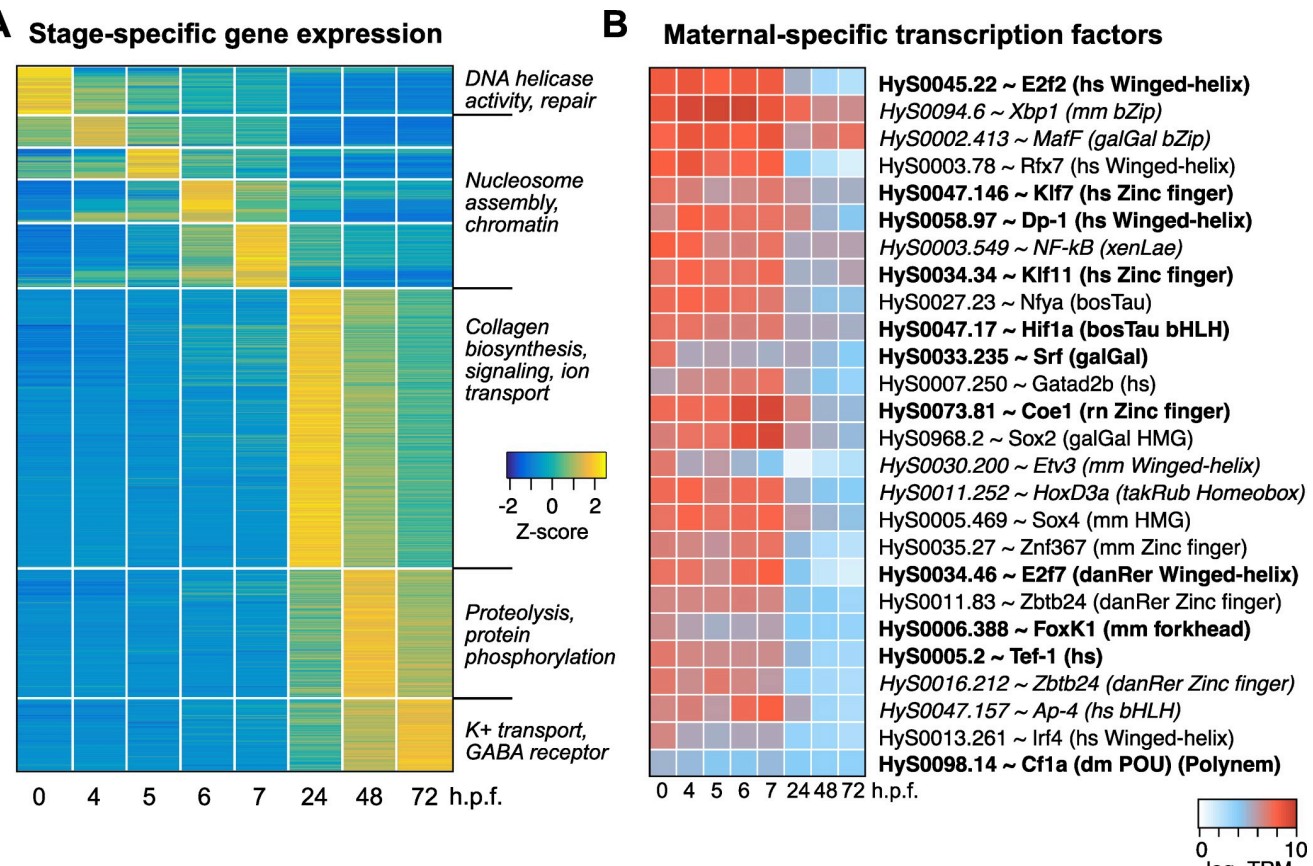

**Fig 4. Stage-specific gene expression is observed throughout the timecourse. (A)** Heatmap showing relative expression levels of genes, grouped according to the time point that expression level peaks. **(B)** Heatmap showing expression levels across time of maternal-specific highly-expressed predicted transcription factors. Each Hydractinia gene is annotated with the best UniProt / SwissProt BLAST hit, species, and DNA binding domain for select factors (hs = human, mm = mouse, galGal = chicken, xenLae = *Xenopus laevis*, bosTau = cow, rn = rat, takRub = Takifugu, danRer = zebrafish, dm = Drosophila). Bold = E < 1x10^{-50}, italics = 1x10^{-25} < E < 1x10^{-10}. TPM = transcripts per million.

[78], is also present in the maternal contribution (HyS0098.14), albeit at lower levels (Fig 4B and S10 Table). Sox2 and POU-homeobox Oct4 homologs play major roles in mammalian pluripotent stem cell induction and genome activation in zebrafish and Xenopus, suggesting that similar factors may help guide Hydractinia genome activation [18,59,79–82].

Another set of transcription factors is specific to the gastrula stages, including the early-activated forkhead-domain containing HyS0062.47, as well as predicted RFX and Atonal family members, which are associated with sensory gene regulation in other animals (S6A Fig and S10 Table). And at 24 h.p.f. and beyond, we observe expression of Homeobox genes and T-Box factors, which likely play roles in morphogenesis (S6A Fig). Together, our RNA-seq timecourse has revealed many potential transcription factors that likely mediate developmental transitions throughout embryogenesis.

## Deadenylation precedes degradation through both transcription-dependent and independent maternal clearance mechanisms

The stage-specific expression profiles show the decline of a subset of maternally contributed mRNA through 7 h.p.f., indicating that Hydractinia embryos undergo maternal clearance concurrently with genome activation similar to other taxa. In all, 5038 genes significantly decrease

at some point from egg stage to 7 h.p.f. (DESeq2 adjusted $P < 0.05$), which is about half of all genes with a maternal contribution (S11 Table).

Downregulation is already observed starting at 2 h.p.f. for 1331 genes in the poly(A)-selected libraries, suggesting that a maternal mode of clearance is initiated shortly after fertilization, independent of genome activation. However, this decline is not observed in the rRNA-depleted libraries until at least 5–6 h.p.f., which indicates that the initial changes are due to poly(A)-tail shortening (Fig 5A–5C). Examination of the maternal contribution indicates that homologs of mRNA encoding all components of the CCR4-NOT complex as well as PARN are maternally provided, which may underlie this early deadenylation activity (S7A Fig).

Additional coordinated groups of mRNA are downregulated at subsequent time points in the poly(A)-selected libraries, but in most cases there is a delay before transcript level decreases are detected in the rRNA-depleted libraries, and rarely before 6 h.p.f. regardless of when deadenylation is first observed (Fig 5A–5C). These observations suggest that targeting and deadenylation factors are maternally provided and act early, but clearance-associated mRNA nucleolytic activity is not triggered until later in embryogenesis.

To test whether genome activation is the trigger for mRNA destabilization, we treated fertilized embryos with the RNA Polymerase II transcription inhibitor Triptolide and compared their transcriptomes to embryos mock treated with DMSO at 7 h.p.f. Triptolide has been shown to inhibit genome activation in other animal embryos similar to α-amanitin [17,83,84], but avoids needing microinjection for delivery. DMSO-treated embryos developed normally, and RNA-seq profiles are highly correlated with untreated embryos (Pearson's r = 0.99, S7B Fig). By contrast, Triptolide-treated embryos fail to properly activate their genomes (S7C Fig), confirming that Triptolide treatment successfully inhibits transcription in Hydractinia embryos.

When we compared clearance profiles in Triptolide versus DMSO-treated embryos, we found that deadenylation was not inhibited in Triptolide for targets of early clearance (initiating 2–4 h.p.f. in untreated embryos), confirming that maternal mechanisms catalyze the initial phase of maternal clearance (Fig 5D, 5E). In fact, there was a trend toward heightened deadenylation in the presence of Triptolide among these mRNA, suggesting that genome activation may actually negatively impact the rate of deadenylation. In contrast, RNA levels as measured in rRNA-depleted libraries were subtly higher ($P = 8 \times 10^{-27}$, Wilcoxon signed-rank test), i.e. stabilized, in the presence of Triptolide, suggesting that genome activation at least partially contributes to mRNA destabilization.

Among mRNA experiencing later maternal clearance (initiating $\geq 5$ h.p.f.), Triptolide treatment inhibited clearance as measured in both poly(A)+ and rRNA-depleted libraries ($P = 5 \times 10^{-90}$ and $2 \times 10^{-165}$, Wilcoxon signed-rank tests), suggesting that these mRNA are cleared in a transcription-dependent manner (Fig 5E). However, as full stabilization was not observed, it is possible that overlapping maternally derived and zygotic mechanisms target these mRNA.

## MicroRNAs and 3'UTR targeting may contribute to maternal clearance

To explore mechanisms underlying maternal clearance in Hydractinia, we investigated properties that have been associated with clearance in other taxa. First, we asked whether codon usage distinguished destabilized versus stable maternal mRNA. In some taxa, cleared mRNA tend to contain sub-optimal codons, which is thought to be associated with poor translation efficiency and ribosome stalling, leading to mRNA destabilization [22,23]. We calculated codon adaptation indices (CAI) using empirically-determined codon frequencies from the most highly expressed maternal mRNA; however, there was no significant difference in CAI

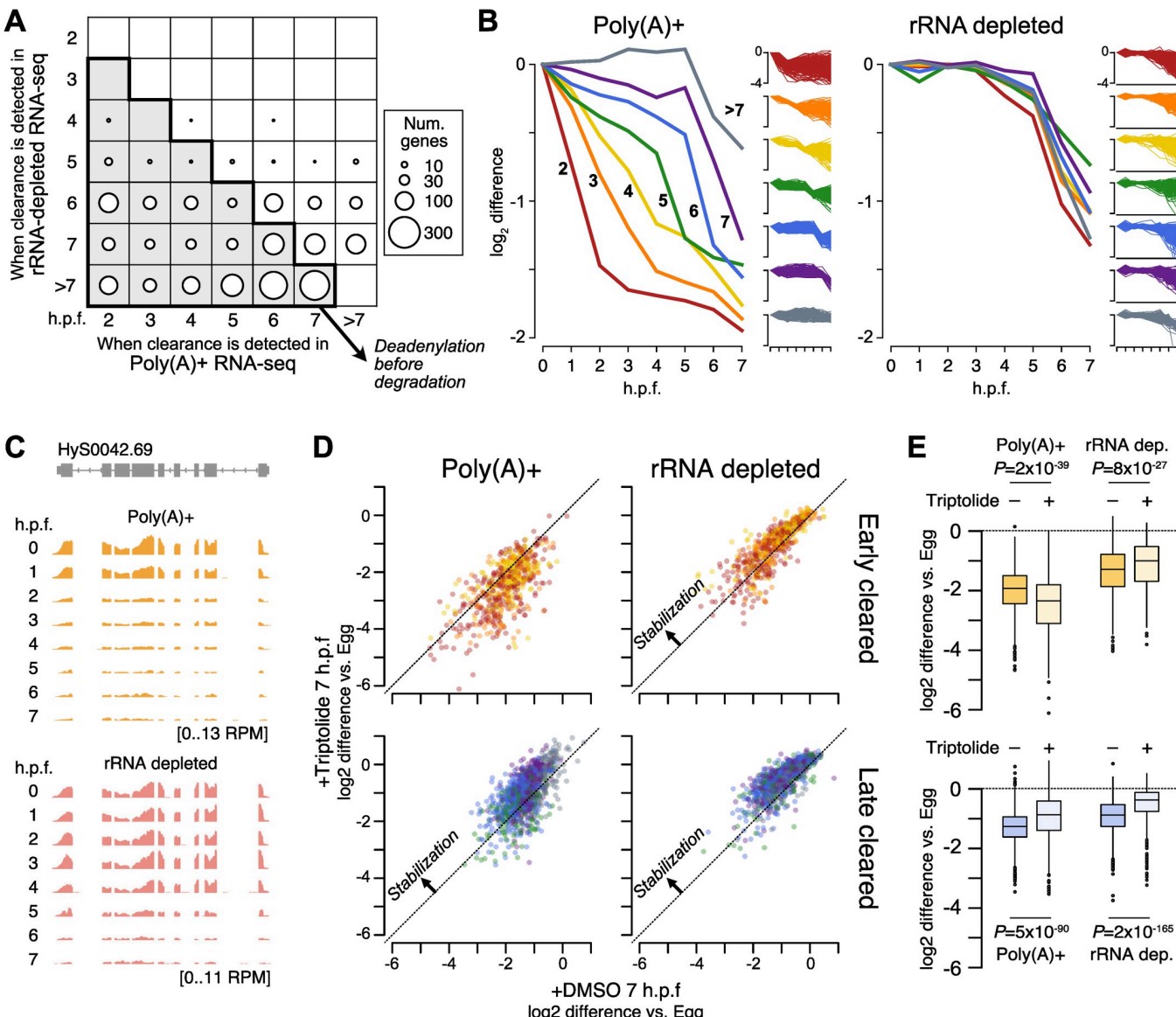

**Fig 5. Maternal mRNA are cleared with different dynamics. (A)** Bubble chart showing the timing of clearance. Each square shows the number of genes with >2-fold expression decrease relative to egg (proportional to size of the circle), according to the time point (h.p.f.) when the decrease is observed in the poly(A) + RNA-seq (x-axis coordinate) versus the rRNA-depleted RNA-seq (y-axis coordinate). The shaded squares are time point combinations where decreases are observed in the poly(A)+ RNA-seq before they are observed in the rRNA-depleted RNA-seq. **(B)** Average profiles of $\log_2$ fold difference relative to egg (0 h.p.f.) of genes grouped according to when expression decreases are observed in the poly(A)+ RNA-seq (2–7+ h.p.f.). Each gene group is plotted for poly(A)+ RNA-seq (left panel) and rRNA-depleted RNA-seq (right panel). Full trajectories for all genes in each group are shown as small insets, 2–7+ h.p.f. from top to bottom. **(C)** Browser track showing asymmetric timing of RNA-seq coverage decreases observed in poly(A)+ RNA-seq (top) versus rRNA-depleted RNA-seq. **(D)** Biplots showing the effect of Triptolide treatment on mRNA clearance as measured using poly(A)+ RNA-seq (left panels) and rRNA-depleted RNA-seq (right panels). Top panels show genes whose clearance was detected 2–4 h.p.f. in poly(A)+ RNA-seq ("Early cleared"), bottom panels ≥5 h.p.f. ("Late cleared"). Genes experiencing equivalent clearance with or without Triptolide treatment would fall on the diagonal. Genes with inhibited clearance would be above the diagonal. **(E)** Quantification of differential clearance with and without Triptolide treatment. Top boxplots are for early-cleared genes, bottom boxplots for late-cleared genes. P values are from Wilcoxon signed-rank tests. h.p.f. = Hours post fertilization.

between cleared and uncleared mRNA, suggesting that codon usage is not a determinant of maternal clearance in Hydractinia (S7D Fig and S1 Table).

Next, we looked for over-represented sequences in the 3' UTRs of cleared mRNA, which could correspond to recognition elements for RBPs. MEME motif discovery analysis [85]

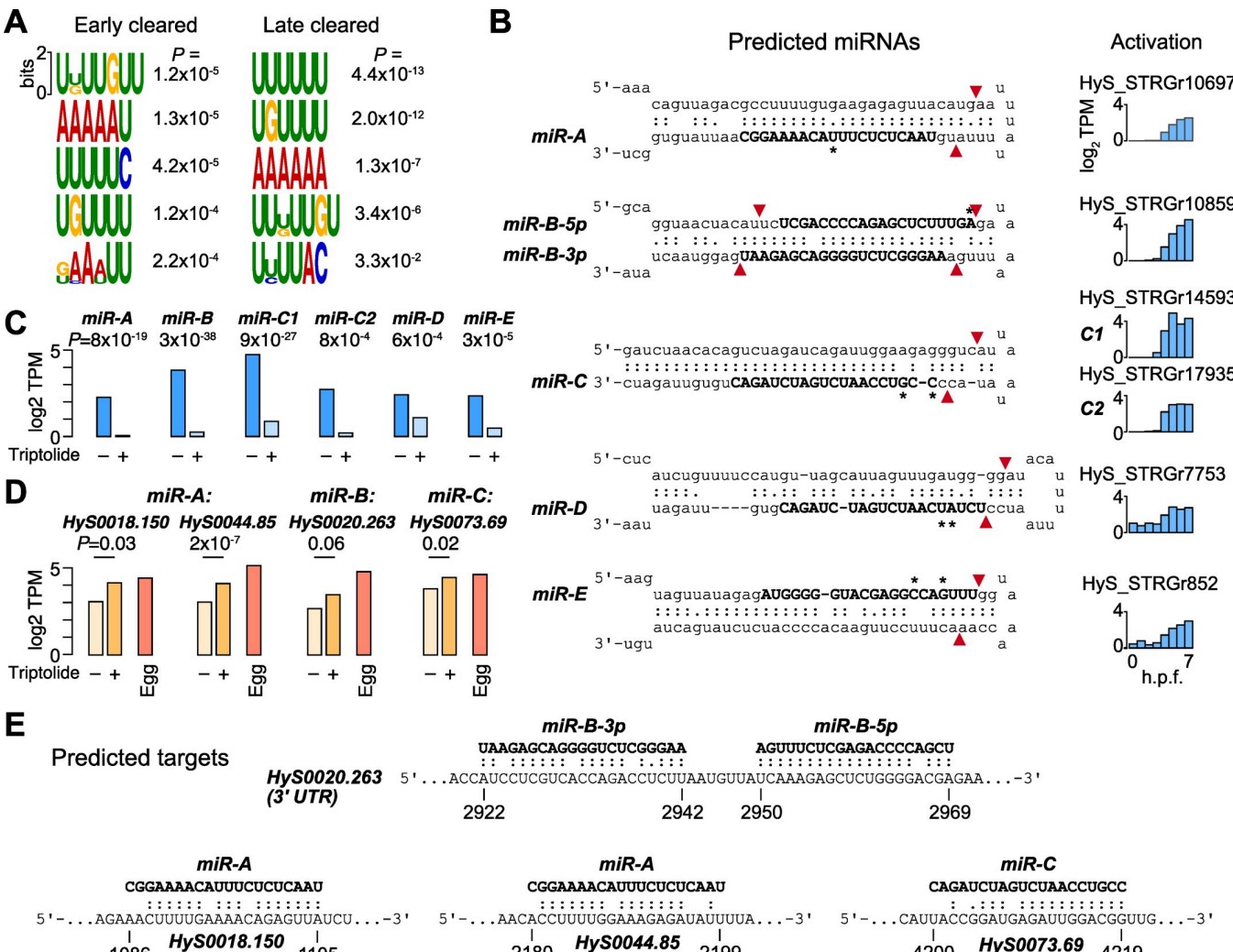

**Fig 6. Clearance may be mediated by RNA-binding proteins and microRNAs. (A)** Motifs found in the 3'UTRs of genes whose clearance is detected 2–4 hours post fertilization (h.p.f.) (left) versus $\geq$ 5 h.p.f. (right). P values are based on enrichment over a background set of stable mRNA, as calculated by the MEME Suite. **(B)** Predicted mature miRNAs (bold) in the precursor hairpin structural context of the primary transcripts that encode them. Red arrows mark predicted Dicer (loop proximal) and Drosha cleavage sites according to canonical processing. Asterisks mark base differences relative to predicted *H. echinata* miRNAs. Expression levels of each primary transcript from 0–7 h.p.f. are shown to the right. **(C)** RNA-seq expression levels of each predicted miRNA primary transcript in the presence or absence of Triptolide. P values are calculated by DESeq2 with FDR adjustment. **(D)** RNA-seq expression levels of predicted mRNA targets of miRNAs in the presence or absence of Triptolide, as compared to maternal levels in the egg. P values are calculated by DESeq2 with FDR adjustment. **(E)** Base-pairing of predicted miRNAs with their predicted mRNA targets. Coordinates are relative to the mature mRNA. TPM = transcript per million.

revealed weak enrichment of A-rich and U-rich motifs in the 3'UTRs of both early- and late-cleared maternal mRNA, when compared to a control set of stable mRNA (Fig 6A). A search for these motifs in the CISBP-RNA Database [86] revealed broad similarity to RRM-domain containing RBPs in other species, suggesting that Hydractinia RRM-containing RBPs participate in maternal clearance (S7E, S7F Fig). We additionally found a homolog of Smaug in the maternal contribution (HyS0016.179), suggesting that it too may have a role in the early phase of clearance (S3 Table), as has been proposed in Nematostella as well [47].

Finally, we considered whether any of the cleared maternal mRNA could be targets of miRNAs. We first interrogated the embryonic transcriptome for components of the miRNA biogenesis and targeting machinery and found that all of the essential enzymes are maternally

provided as mRNA (S8A Fig), suggesting that the embryo is competent for miRNA-mediated regulation.

We then looked for evidence of primary miRNA transcripts in the embryonic transcriptome. We obtained *Hydractinia echinata* mature miRNA sequences from the Hydractinia genome sequencing project and Nematostella and Hydra miRNAs from miRBase [87], then performed BLAST alignment against our Stringtie de novo RNA-seq transcriptome. Of the 103 hits with E-value < 0.1, we filtered for significantly activated transcripts (DESeq adjusted *P* < 0.05) predicted by Vienna RNAfold [88] to adopt RNA hairpin secondary structures typical of miRNA precursors. This yielded six candidate miRNAs, which we call here *miR-A*, *miR-B-5p*, *miR-B-3p*, *miR-C*, *miR-D*, and *miR-E*, each homologous to *H. echinata* miRNAs with 1–2 mismatches, but not the other cnidarian miRNAs (Figs 6B and S8B, S8C and S9 Table). *miR-B-5p* and *miR-B-3p* map to opposite ends of the same putative precursor hairpin, slightly offset compared to the typical 2-nt overhangs induced by Drosha and Dicer cleavage, though variations in duplex position do occur in other miRNAs [89]. *miR-A* also has a noncanonical position and size relative to the predicted Dicer cleavage product; however, its *H. echinata* homolog is the most strongly scoring predicted miRNA, supported by >250,000 small RNA-seq reads [90]. The *miR-C* sequence was found duplicated in two different predicted genes (C1 and C2) in long stem loops uncharacteristic of Drosha processing [91] (S8D Fig), but instead resembling endogenous siRNA substrates that have been identified in other animals [92,93]. Expression of all the predicted miRNA primary transcripts is inhibited in Triptolide-treated embryos, confirming that they are zygotically encoded genes (Fig 6C), though there also appears to be a small maternal contribution as well (e.g., *miR-D*) (Fig 6B).

To identify potential targets of the candidate miRNAs, we performed BLAST alignment against cleared maternal mRNA sequences, anticipating extensive sequence complementarity between the mature miRNA and the mRNA consistent with what has been observed for other cnidarian miRNAs [94]. We recovered nine mRNA with strong antisense complementarity to one or more of the miRNA sequences (Figs 6D, 6E and S8E–S8G). No potential target sequences were found among the stable mRNA control set. Of these, three had significantly higher levels in Triptolide-treated embryos (DESeq2 adjusted *P* < 0.05), consistent with the inhibition of their predicted miRNA regulators (Fig 6D, 6E). Beyond these three, the cleared mRNA HyS0020.263, encoding a predicted glutamate metabotropic receptor, appears to have adjacent target sites in its 3'UTR to both *hsy-miR-B-5p* and *hsy-miR-B-3p*, but only weak stabilization in Triptolide-treated embryos (DESeq2 adjusted *P* = 0.06) (Fig 6D, 6E). Although further experimental verification of both the predicted miRNAs and targets is needed, along with a more comprehensive survey of embryonic small RNAs, these results suggest miRNAs play roles alongside other factors in Hydractinia maternal clearance.

## Discussion

In sum, our transcriptome profile of early *H. symbiolongicarpus* embryogenesis has revealed strong thematic similarities in the maternal-to-zygotic transition between a cnidarian and bilaterians (Fig 7). Embryonic genome activation is preceded by a period of transcriptional quiescence, during which maternally contributed mRNA are either up-regulated via readenylation, or down-regulated via deadenylation as part of the first phase of maternal clearance. After genome activation, subsequent phases of maternal clearance are triggered, potentially involving de novo transcribed miRNAs. Thus, cytoplasmic polyadenylation, delayed genome activation, and maternal mRNA clearance are shared features of the MZT in both non-bilaterian and bilaterian animals. However, evidence for unusual embryonic histone and chromatin

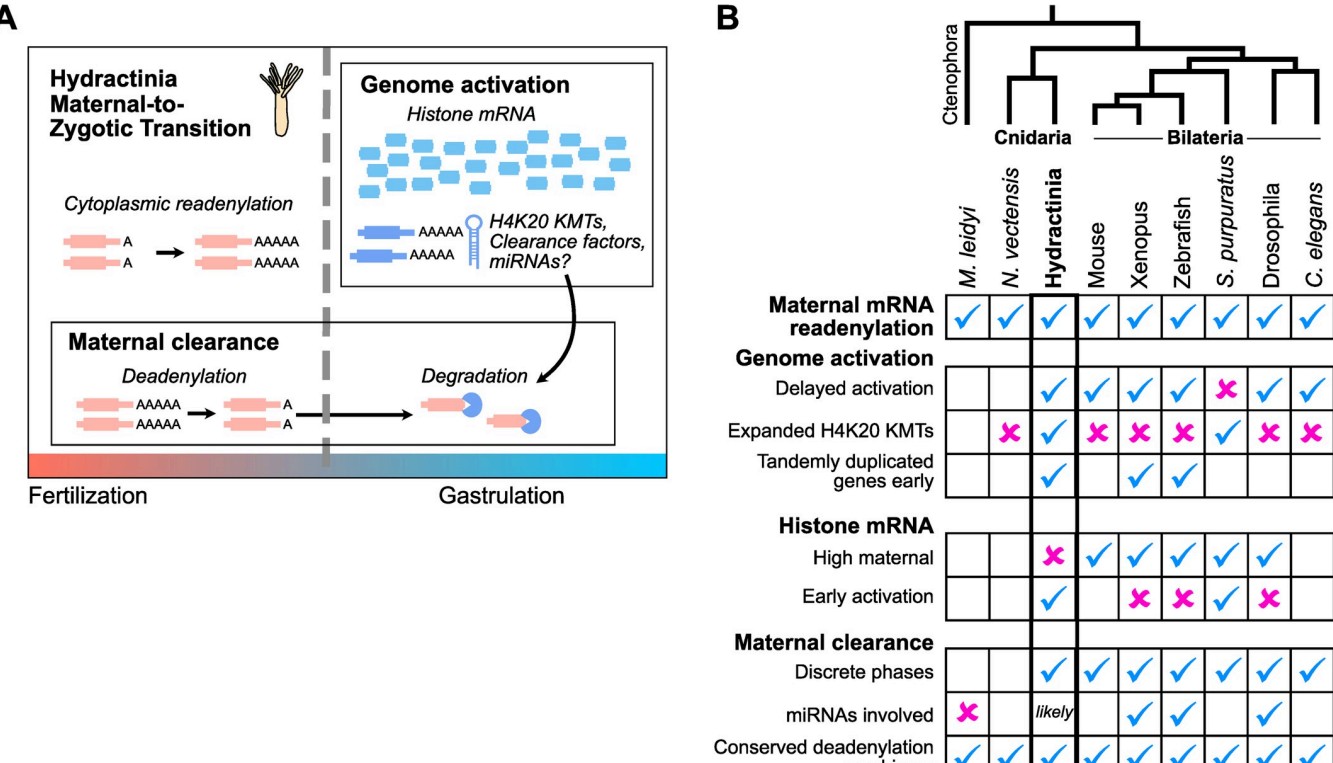

**Fig 7. The maternal-to-zygotic transition in Hydractinia compared to other taxa. (A)** In Hydractinia, prior to genome activation, maternal mRNA are subject to either poly(A) tail lengthening, or shortening as part of the maternal clearance program. After genome activation, histone mRNA predominate in the transcriptome, and an expanded repertoire of H4K20 methyltransferases as well as unknown clearance factors and predicted miRNAs are also de novo transcribed. These clearance factors likely initiate the mRNA degradation phase of maternal clearance. **(B)** Table comparing features of the Hydractinia maternal-to-zygotic transition shared or not shared with other taxa. A blue checkmark indicates the feature applies to the taxon, magenta "x" indicates it does not apply, empty boxes indicate undetermined or not definitively known–e.g., *M. leidyi* and *N. vectensis* may have delayed genome activation, but sufficient temporal resolution is currently lacking to determine this for sure. Evolutionary relationships between the taxa are represented by the cladogram above, with the Ctenophora and Cnidaria phyla and the Bilateria clade labeled.

regulation likely involving H4K20 methylation so far distinguishes the Hydractinia MZT and warrants further investigation.

Our observations were enabled by our rRNA-depletion strategy for RNA-seq, which allowed us to distinguish changes in absolute RNA levels from changes in poly(A) tail length manifested as differential pulldown efficiency when performing poly(A)+ RNA-seq [49]. Comparison between poly(A)+ and rRNA-depleted RNA-seq facilitated our discovery of cytoplasmic polyadenylation in a cnidarian embryo, which was recently demonstrated in Nematostella and *M. leidyi* as well [95], and thus is likely a common feature across embryos in the animal tree (Fig 7B). rRNA-depleted RNA-seq also allowed us to dissect the deadenylation dynamics of maternal clearance (Fig 5), as well as accurately quantify non-adenylated histone mRNA, introns from pre-mRNA, and non-coding RNA (Figs 1, 2 and S3). In the future, pairing rRNA-depletion with metabolic labeling / nascent RNA-seq strategies could increase detection sensitivity even further by selecting for rare copies of newly synthesized pre-mRNA from an embryonic transcriptome dominated by the maternal contribution [57,58].

Genome activation in Hydractinia is widespread across thousands of genes that were also expressed in the maternal contribution (Fig 1B–1D), similar to other animals [2,18,31,48,59]; as well as repetitive and transposon-associated elements, reinforcing a growing trend of the roles of transposons in shaping early embryonic gene expression across taxa by contributing

regulatory sequences that can be co-opted to drive early gene activation [69,75–77]. However, it is the massive activation of histone gene transcription that underlies the major change to the blastula/gastrula transcriptome (Fig 2). The genomic configuration of these histone genes, which are short and encoded in long tandemly repeated arrays, may facilitate their early activation. Shorter genes tend to be activated earlier across different taxa's embryos, for which initially rapid cell cycles present a conflict between DNA replication and transcription machinery that may be unfavorable for longer genes until the cell cycle slows [58]. In zebrafish and Xenopus, the repetitive organization of miRNA genes may also explain why they are the first to be activated in those species. The high concentration of promoters in the ~500 kb zebrafish miR-430-encoding locus was found to form a transcription body during early embryonic genome activation, which may subsequently spread to other genes [17,96,97]. Perhaps this genomic niche is served instead by the large histone gene arrays in Hydractinia. MiRNAs and histones play very different roles during the MZT, but they may share a requirement to be rapidly and abundantly transcribed in their respective early embryos.

Hydractinia embryos have an extremely low maternal histone mRNA concentration relative to other characterized taxa, which increases 18-fold after genome activation to dominate the embryonic transcriptome (Fig 2C). We found that this is in contrast to zebrafish, Xenopus, and Drosophila, where the major increase in histone mRNA production occurs later, to presumably meet the chromatin needs of thousands of newly synthesized genome copies; and mouse eggs were reported to also have a high maternal histone mRNA contribution that rapidly declines after the first cleavage [89]. Conversely, maternally derived high histone concentrations are inhibitory to early gene activation in many animals [9–13], so embryos likely strike a balance between histone and DNA concentration that helps facilitate the timing of gene expression in the context of rapid cell division during the MZT.

How is this histone balance achieved in Hydractinia embryos? One possibility is that histone concentration is not used as a barrier to genome activation in this taxon, but its sharp increase after genome activation could facilitate a shift toward repressive chromatin in the gastrula, perhaps in service of restricting potential in rapidly differentiating cells. Indeed, the co-expression of a large number of predicted H4K20 methyltransferases at genome activation (Fig 3) could be correlated with this chromatin remodeling activity, and the concomitant clearance of a maternal H4K20 methyltransferase variant (Fig 3A) suggests a maternal-to-zygotic chromatin regulatory transition for this mark. H4K20 methylation has been implicated in differential chromatin accessibility [72,73], and thus may participate in ensuring the activation of only specific transcriptional programs in embryonic cells. The potential for a similar expansion of H4K20 methyltransferase genes in sea urchin embryos (S4 Fig) raises the possibility that this represents an ancient mode of chromatin regulation that was only selectively maintained in certain animals, for reasons that remain to be discovered (Fig 7B). Of course, RNA-seq does not capture protein concentrations or translation dynamics, which may influence the interpretation of histone mRNA levels relative to chromatin states.

Phenomena other than histone-mediated repression would likely underlie genome activation timing in Hydractinia, many of which have been characterized in other taxa [1]. An intriguing phenomenon recently identified in Hydractinia suggests that the DNA base modification 6-methyladenine (6mA) plays a role, such that its random incorporation in the early embryonic genome inhibits gene activation [53].

The factors guiding Hydractinia genome activation also remain to be discovered among the many maternally provided transcription factor mRNA (Fig 4B); however, notable in the maternal contribution was a predicted HMG-domain factor with similarity to vertebrate Sox2 and the POU-domain containing Polynem, previously associated with Hydractinia stem-cell induction [78]. Zebrafish Pou5f3 (Oct4 homolog) and Sox19b (Sox2 homolog) along with

Nanog, and Xenopus Pou5f3 and Sox3 (Sox2 homolog) are also maternally provided and required for genome activation in their respective species [18,59,79–81]. POU-domain and HMG-domain factors are found in the ctenophore *M. leidyi* maternal contribution as well [41]. Together, this suggests that homologs of mammalian pluripotency factors Oct4 and Sox2 may play ancient roles in animal embryonic stem cell induction. However, as not all animals use these factors for genome activation–notably Drosophila, which rely on the unrelated factors Zelda, CLAMP and GAF [98–101]–it is also possible that novel maternal transcription factors will also emerge as Hydractinia genome activators.

We find that at least half of the maternal contribution is targeted for clearance (Fig 5), which is similar in scale to bilaterian embryos [1]. Clearance is initiated through deadenylation by transcription-independent mechanisms, indicating that Hydractinia embryos have a maternal mode of clearance. The identification of PARN and CCR4-NOT components in the Hydractinia maternal contribution, along with similar findings in Nematostella [47] and *M. leidyi* [41], reinforces the roles of deeply conserved deadenylation pathways in mediating maternal clearance across animals [1,47] (Fig 7B). However, nucleolytic degradation of these transcripts does not occur until after genome activation, suggesting that additional zygotic clearance factors must be de novo transcribed. Clearance targets were not fully stabilized upon Triptolide treatment, which could be due to incomplete efficacy of the drug (S7C Fig) as has been previously reported [102], or could indicate overlapping, redundant maternal and zygotic mechanisms that contribute to mRNA clearance. Parallel maternal and zygotic modes of clearance are found across bilaterians [23,67,103,104], and our findings establish this phenomenon in cnidarians as well. We also note that Triptolide may affect other cellular components, which may or may not be downstream consequences of transcription inhibition [102], so it is possible that these could have an impact on mRNA stability. Triptolide has also been reported to activate the calcium channel Polycystin-2 [105], which could affect intracellular $Ca^{2+}$ levels in the embryo, and there has been at least one report linking calcium signaling with mRNA half life [106]. The use of alternative inhibitors such as α-amanitin in the future would help clarify the specificity of Triptolide in Hydractinia embryos.

Finally, we predict that a select number of maternal mRNA are cleared by de novo transcribed miRNAs (Fig 6). Direct quantification of *H. symbiolongicarpus* embryonic miRNAs remains to be done; however, there is strong evidence that primary miRNA transcripts are among the de novo activated genes in the embryo by sequence matches to *H. echinata* miRNAs (S8B Fig). We cannot exclude the possibility that maternally provided miRNAs also exist, though given that cnidarian miRNAs induce cleavage rather than deadenylation [94], which we do not observe until after genome activations, unknown mechanisms would likely delay their activity initially. Although miRNAs play major roles in the zygotic mode of maternal clearance in zebrafish [35,37] Xenopus [38], and Drosophila [36], a comparable scope of miRNA-mediated clearance was unlikely to exist in Cnidarians: because bilaterian miRNAs canonically recognize mRNA via short 6–8 bp regions of complementarity in 3' UTRs, they can target hundreds of maternal mRNA [91]. In contrast, cnidarian miRNAs have been found to rely on full-length complementarity between the ~22 nt miRNA and the target mRNA [94], which would tend to limit miRNA regulation to only one or a few mRNA. Thus, the existence of miRNA-mediated maternal clearance in a cnidarian embryo likely reflects convergent evolution of a post-transcriptional regulatory strategy during the MZT. Indeed, clearance-associated Drosophila miR-309 and zebrafish/Xenopus miR-430/427 are unrelated [1], suggesting that independently evolved deployment of core gene regulatory mechanisms is a common feature of the MZT across taxa.

In conclusion, our dissection of the maternal-to-zygotic transition in a cnidarian fills a long-standing gap in our goal to understand the unifying features across animals of early

embryonic gene regulation. Although we continually find that the mechanistic details vary widely between species, shared gene regulatory themes emerge. This motivates the further inclusion of embryos from a larger collection of taxa, to uncover the ancient regulatory logic that may underlie early animal development, and better understand the context in which species-specific strategies have evolved.

## Methods

### Embryo collection

*H. symbiolongicarpus* colonies were maintained and spawned as described in [107]. Gametes were collected from breeder colonies 291–10 and 295–8 and used for fertilization in artificial sea water (Instant Ocean Reef Crystals ~28–31 ppt) at 20˚C in a petri dish, reserving a portion of unfertilized eggs for immediate collection. After 15 minutes, fertilized eggs were washed, then the first time point was collected 15 minutes later while still at 1-cell stage (30 min post fertilization, m.p.f.). Embryos were transferred to a 23˚C incubator for subsequent collections. For each collection, approximately 100 embryos were transferred into a 1.5 ml eppendorf tube, excess water removed, flash frozen in liquid nitrogen, and stored at -80˚C until RNA extraction.

For Triptolide (Apexbio #MFCD00210565) treatment, a 4 mM stock solution dissolved in DMSO was added to each treated embryo dish at 30 m.p.f. to a final concentration of 20 μM Triptolide and 0.5% DMSO in seawater. DMSO was added in parallel to mock-treatment embryos dishes to 0.5% DMSO final. Batch-matched untreated embryos in seawater were kept for reference. Embryos were collected at 7 h.p.f.

### RNA extraction

Frozen embryos in 1.5 ml eppendorf tubes were homogenized with a pestle in 500 μl of TRIzol (Invitrogen #15596026) (note that for later timepoints 24–72 h.p.f., some particulates remained after pestling). 100μl chloroform was added, vortexed for 15 sec, then centrifuged at 13,000 RPM at 4˚C for 15 minutes. The aqueous phase plus 340 μl of isopropanol and 1 μl of GlycoBlue (Invitrogen #AM9515) were added to a new tube and precipitated overnight at -20˚C. To pellet, tubes were centrifuged at 13,000 RPM at 4˚C for 30 minutes, supernatant removed, 500 μl of cold 75% ethanol added, and centrifuged at 13,000 RPM at 4˚C for 15 minutes. Ethanol was drawn off with a P200, then P10 pipet and tubes left to air dry in a fume hood for 10 minutes. Pellets were resuspended in 50 μl of nuclease-free water, concentration measured by NanoDrop, and stored at -80˚C until use.

### rRNA antisense oligo design

rRNA content in the early embryo was assessed by pilot RNA-seq experiments on 2-cell *H. symbiolongicarpus* embryos (0.5–1 h.p.f. at 20˚C), building parallel total-RNA and poly(A) + Illumina strand-specific libraries using the Ultra II RNA-seq library kit (NEB #E7765) (see below). Total RNA-seq reads were used for Trinity v2.12.0 de novo transcriptome reconstruction [108] to empirically determine the specific rRNA species represented in the embryonic transcriptome, which recovered two nearly identical 5–6 kb contigs corresponding to the 45S pre-rRNA locus. Refseq-annotated *N. vectensis* 18S, 5.8S, and 28S rRNA sequences aligned the contig at expected positions (XR_004291954, XR_004294136, and XR_004293519, respectively). BLAST alignment of the de novo assembled 45S rRNA to the genome assembly revealed multiple full-length and partial hits across the Hsym 1.0 genome assembly, including 10 tandem copies on scaffold HyS0316 (S1D Fig).

To identify expressed mitochondrial rRNA, total RNA-seq reads were aligned using bowtie2 [109] to a Genbank-deposited *H. symbiolongicarpus* full mitochondrial genome sequence (LN901197.1), and coverage was confirmed over the 16S and 12S rRNA encoding regions. These regions in turn align to the genomic scaffold HyS0613. In all, <1% of all aligned reads corresponded to the mitochondrial rRNA in total rRNA libraries, compared to 97% aligned to 45S rRNA.

The predicted nuclear (28S, 18S, and 5.8S) and mitochondrial (16S, 12S) rRNA sequences were used as inputs to our Oligo-ASST web tool [49], generating 136 39-40mers spaced on average 20 bp apart that tiled the five rRNA genes (S2 Table). Oligos were ordered from Thermo Fisher as individual dry, desalted tubes at 25 nmol scale, resuspended to 1000 µM with nuclease free water, then separate pools were created for the nuclear and mitochondrial oligos by combing 1 µl of each tube and diluting to a final 10X stock concentration of 4 µM per oligo for nuclear, 1 µM per oligo for mitochondrial, as per [49].

## RNA-seq library construction

Each biological sample from the early time points (0–7 h.p.f.) was split for two different RNA-seq preps, poly(A)+ selected and rRNA-depleted; 24–72 h.p.f. samples were subjected only to poly(A)+ prep.

rRNA depletion proceeded as described in [49]. 1 µl of each of the two 10X rRNA depletion oligo stocks was combined with 1 µg of total RNA in a 10µl hybridization reaction (100 mM Tris–HCl pH 7.4, 200 mM NaCl, 10 mM DTT) in a 0.2 µl PCR tube, to a final concentration of 0.4 µM per nuclear oligo and 0.1 µM per mitochondrial oligo. The reaction was heated in a thermal cycler at 95˚C for 2 min, then cooled to 22˚C at a rate of -0.1˚ C / sec and held for 2 min. Digestion was performed with 2 µl (10U) of thermostable RNaseH (NEB #M0523S) and 2 µl of 10X buffer in a 20 µl volume at 65˚ C for 5 min, then placed on ice. 2.5 µl of TURBO DNase (Invitrogen #AM2238) with 5 µl of 10X buffer was added, volume brought to 50 µl, then incubated at 37˚C for 30 min. The RNA was size-selected to ≥200 nts using the Zymo RNA Clean and Concentrator-5 kit (Zymo #R1013) according to manufacturer's protocol and eluted in 6 µl of nuclease-free water.

Poly(A) selection was performed using NEB-Next Poly(A) mRNA Magnetic Isolation Module (NEB #E7490L) according to the manufacturer's protocol starting with 1 µg of total RNA and performing two rounds of selection. RNA was eluted off of the beads with provided Tris buffer, then cleaned up with the Zymo kit with in-column DNase treatment and eluted in 6 µl of nuclease-free water.

Strand-specific RNA-seq libraries were constructed using the NEB Ultra II RNA-seq library kit (NEB #E7765) according to manufacturer's protocol with the following parameters: 12–35 ng of rRNA-depleted or poly(A)+ selected RNA was fragmented at 94˚ C for 15 minutes in first-strand buffer, then subjected to half-reaction volume library prep with 5-fold dilution of adapters and 9 cycles of PCR amplification; Sera-Mag Select beads (Cytiva 29343045) were used for cleanup steps. Libraries were quantified using Qubit dsDNA high sensitivity (Invitrogen #Q32851) and sequenced paired end on an Illumina NextSeq 500 or NextSeq 2000 at the Health Sciences Sequencing Core at Children's Hospital of Pittsburgh.

## RNA-seq data analysis

RNA-seq reads were trimmed of trailing Ns and adapter sequence using Trim Galore v0.6.6 (https://github.com/FelixKrueger/TrimGalore) with Cutadapt v1.15 [110]. For gene-level quantification, RNA-seq reads were pseudo-aligned to the Hsym 1.0 protein-coding transcriptome [90] using kallisto v0.46.1 [111] in paired rf-stranded mode. Counts were rounded to

integers. Differential expression analysis was performed using DESeq2 v4.0.3 [112] on poly(A) + samples and rRNA-depleted samples separately, discarding genes that had <10 reads across all samples. Genes with FDR-adjusted DESeq2 *P* value < 0.05 between two samples were considered differentially expressed. An expression threshold of 1 TPM was used to distinguish evidence of any gene expression, while 5 TPM was used to define gene subsets with robust expression.

For de novo transcriptome prediction, RNA-seq reads were first aligned to the Hsym 1.0 genome sequence using HISAT2 [113] with the—rna-strandness RF parameter. Stringtie v2.2.1 [114] was used for step-wise construction of an augmented transcriptome annotation. First, two poly(A)+ Stringtie transcriptomes were constructed separately using poly(A)+ libraries from 3–7 h.p.f. and libraries from 24–72 h.p.f. For each set, Stringtie was run with minimum transcript length = 100 bp and minimum proportion (-f) = 0.25 using the Hsym v1.0 protein-coding gene annotation as a reference GTF. The two poly(A)+ Stringtie transcriptomes were merged using stringtie–merge, then the GTF file was filtered to exclude any chimeras of Hsym v1.0 annotated genes; for each chimeric Stringtie gene removed, the original Hsym v1.0 gene annotations involved in the chimera were added back. Next, a Stringtie transcriptome was constructed using rRNA-depleted libraries from 3–7 h.p.f. and compared with the poly(A)+ Stringtie transcriptome GTF using gffcompare v0.12.6 [115]. rRNA-depleted Stringtie genes overlapping poly(A)+ Stringtie genes were filtered out (keeping only codes x, i, y, p, and u), and the remaining genes were combined with the poly(A)+ Stringtie transcriptome. Gene IDs have the prefix HyS_STRGp for annotations based on the poly(A)+ libraries and HyS_STRGr for annotations based on the rRNA-depleted libraries.

For gene count quantification on the combined Stringtie transcriptome, kallisto was used with a filtered set of transcript sequences that excluded certain non-coding RNA and repetitive sequences (see below, Gene annotation).

For intron-sensitive read count quantification, the unfiltered Stringtie transcriptome was used with featureCounts v2.0.1 [116] in reverse-strand paired mode on the HISAT2-aligned reads for rRNA-depleted samples, first using transcript features (-t transcript) to get pre-mRNA counts, then to exonic features (default) to get typical mRNA counts. Only unique, unambiguously mapping reads are counted, and only genes corresponding to an original Hsym v1.0 annotated gene were retained (the entire Stringtie-augmented transcriptome was used for quantification to ignore intronic reads that could ambiguously be assigned to novel genes). An intron count table was constructed by subtracting exon counts from pre-mRNA counts, then filtered to exclude genes whose intronic reads per kilobase per million (RPKM) was less than 0.5 (summed intronic length was used for the length normalization factor, summed pre-mRNA counts across all genes was used for the library normalization factor). The exonic and intronic counts were combined into one counts table for DESeq2 differential expression, such that each gene could have up to two different entries for exonic and intronic counts, respectively. Significantly up-expressed genes based on intron signal would have adjusted P < 0.05 for the gene's intron entry.

For the intron RNA-seq coverage heatmap and metaplots, a BED file of introns was constructed from the Hsym v1.0 gene annotations, removing any introns that overlapped any exon in the Stringtie-augmented transcriptome. This BED file was used with deeptools v3.5.1 [117] computematrix scale-regions command with +/- 100 bp flanking (exonic). For the windowed genome activation analysis, log2 fold difference in 10kb windows across the genome was calculated using deeptools bigwigCompare between two bigWig coverage files from HISAT2 alignment according to the comparison.

Additional analyses were performed using R and python3.

## Gene annotation

Hsym v1.0 amino acid sequences were BLASTed against the SwissProt database, downloaded from NCBI on Feb 4, 2023 using ncbi-blast+ v2.11.0 [118], retaining the best hit per gene. Gene Ontology annotations keyed to the SwissProt accession numbers were downloaded from http.ebi.ac.uk/pub/databases/GO/goa/UNIPROT on Mar 18, 2023. GO terms were associated with Hsym genes according to the terms annotated for the best SwissProt BLAST hit, only if the BLAST match had E < 1e-5.

For the novel StringTie-predicted genes (non-overlapping with existing gene annotations), an initial blastn search against Nematostella and Hydra non-coding RNAs curated from RNA-central [119] (E < 1e-5, word_size 11) identified the abundant, mostly Pol-III transcribed non-coding RNAs, which were excluded from subsequent annotation. Hits annotated as "FAK," "microRNA," "uncharacterized," or "phosphatidylinositol" were not excluded. The remaining genes were scanned against the DFAM curated repeat element database [120] using HMMER v3.1b2 [121] (hmmscan -E 1e-3—noali—notextw); annotated for low-complexity sequence using Dust [85]; blastx (translated) searched against SwissProt (E < 1e-3); and blastn searched against the RefSeq nucleotide database, downloaded from NCBI on Feb 6, 2023 (E < 1e-5, word_size 11), retaining the best hit for each search. Genes were clustered according to sequence similarity by CD-HIT [122] (≥50 nts of >80% sequence identity). Maximum open-reading frame length was calculated using EMBOSS sixpack v6.6.0.0 [123].

For histone annotation, *H. echinata* histone protein sequences reported in [60] were downloaded from Genbank (accession numbers KX622123-KX622141) and used for blastp searches querying Hsym genes, retaining the best hit per gene. Conserved gene structure between *H. symbiolongicarpus* and *H. echinata* was used to assign specific gene names for histone paralogs with indistinguishable amino acid sequences.

The NCBI web implementation of CD-Search Domain analysis [124] was used to annotate the predicted Hydractinia H4K20 methyltransferase protein sequences. To identify other species' putative H4K20 methyltransferases, *Hydra vulgaris* (Hydra 2.0 Augustus protein models), *Nematostella vectensis* (jaNemVect1.1 NCBI Annotation Release 101), *Mnemiopsis leidyi* (reference and unfiltered models from the Mnemiopsis Genome Project Portal https://research.nhgri.nih.gov/mnemiopsis/), and *Strongylocentrotus purpuratus* (Spur5.0) protein sequences were searched against the CDD NCBI database, downloaded on Jun 14, 2023 using rpsblast with parameters -max_target_seqs 5 -evalue 0.1.

For RNA-seq read count quantification on the Stringtie transcriptome, certain non-coding genes were excluded based on the above annotations: all ncRNA from the cnidarian RNAcentral search, additional tRNA, U7, and U17 RNAs identified by Dfam search, spliceosomal RNA identified by the RefSeq search, any additional novel genes clustered with these ncRNA by CD-HIT (guilt by association), and any predicted genes with >50% of their length low complexity sequence according to Dust analysis.

## Stage-specific expression analysis

Stage-specific genes were identified as having <1 TPM average expression in at least one stage and >5 TPM average expression in another stage and significantly different (DESeq2 adjusted P < 0.05) in any stage compared to egg, based on poly(A)+ samples. Genes were clustered according to the stage when highest expression is observed.

GO enrichment analysis was performed by counting occurrences of each GO term associated with the genes in each stage cluster, pooling the 4–7 h.p.f. clusters due to low gene counts. For each GO term, a 2x5 Chi-squared test was performed (# genes annotated with the GO

term, # genes not annotated; for 5 stage clusters) (991 tests). P values were FDR adjusted using the Benjamini-Hochberg method as implemented in R.

For transcription factor stage expression comparisons, genes annotated with the GO terms GO:0003700, DNA-binding transcription factor activity; and GO:0043565, sequence-specific DNA binding were extracted and manually curated to retain transcription factors. Maternal transcription factors of interest had >50 TPM expression in an egg rRNA-depleted sample, except for Polynem, which was manually included. The most strongly maternal factors were shown in the heatmap (highest egg expression of the genes with >5-fold enrichment in egg compared to some other stage, except for Polynem). Later-stage-specific transcription factors were obtained by intersecting the stage-specific gene list with the transcription factor list.

## Cross-species histone expression quantification

For Drosophila, RefSeq coding RNA annotations were downloaded from the UCSC genome browser on Feb 23, 2023, and histone genes were identified as annotated (His1/His2A/His2B/His3/His4 name prefixes, plus BigH1 and cid). Zebrafish Ensembl r109, Xenbase *X. tropicalis* v10.0, and Echinobase sp5_0 *S. purpuratus* coding regions were BLASTed against SwissProt to identify the histone genes in these species. kallisto gene count quantification for Drosophila, zebrafish, and Xenopus was done for the following samples downloaded from public repositories as follows.

Drosophila (NCBI SRA SRP068959, reverse stranded and single [64], and ENA E-MTAB-11580, rf stranded and paired [63]): SRR3129014 (egg), ERR9389553 (95 m), ERR9389557 (105 m), ERR9389561 (115 m), ERR9389564 (125 m), ERR9389567 (145 m), ERR9389570 (160 m), ERR9389571 (175 m), ERR9389576 (190 m), ERR9389581 (220 m).

Zebrafish (NCBI SRA SRP189512 and SRP149556, rf stranded and paired [67]): SRR8788633 (1-cell), SRR5893050 (64-cell), SRR8788673 (256-cell), SRR5893043 (1k-cell), SRR8788676 (oblong), SRR8788679 (sphere), SRR8788695 (dome), SRR8788699 (50% epiboly), SRR5893169 (shield), SRR8788656 (75% epiboly).

*X. tropicalis* (NCBI SRA SRP053406, rf stranded and paired [66]): SRR1795631 (0 h.p.f.), SRR1795666 (2 h.p.f.), SRR1795676 (3.5 h.p.f.), SRR1795678 (4.5 h.p.f.), SRR1795632 (5 h.p.f.), SRR1795634 (5.5 h.p.f.), SRR1795636 (6.5 h.p.f.), SRR1795638 (7.5 h.p.f.), SRR1795640 (8.5 h.p.f.), SRR1795642 (9.5 h.p.f.), SRR1795645 (10.5 h.p.f.).

The *S. purpuratus* RNA-seq is CAGE-seq data, so bowtie2 alignment on the transcriptome was performed instead on the following samples (NCBI SRA SRP311353, forward stranded and single [65]): SRR14009432 (0 h.p.f.), SRR14009433 (6 h.p.f.), SRR14009434 (12 h.p.f.), SRR14009435 (18 h.p.f.), SRR14009436 (24 h.p.f.), SRR14009437 (30 h.p.f.).

Expression level increase at each time point was calculated as the reads-per-million difference compared to the maximum of the first two time points, which are pre-genome activation (except for *S. purpuratus*, for which only 0 h.p.f. was used as time 0), summed across all genes per category (histone or non histone) that have at least a 2-fold increase. For *S. purpuratus*, a trimmed total read sum was used for normalization (genes in the 10th-90th percentile) to prevent the high histone counts from skewing RPM values.

The zebrafish mir-430 region sequence was extracted from GRCz11 chr4:28694328–28710737 and the *X. tropicalis* miR-427 regions were extracted from XENTR_10.0 Chr3:146283012–146364041 (+ strand cluster) and Chr3:146258923–146283126 (- strand cluster). Total read count to the entire region per sample per species was obtained by bowtie2 alignment.

## Analysis on cleared mRNA

To define clearance groups, genes significantly decreased (DESeq2 adjusted P < 0.05) compared to egg stage were identified at each subsequent time point, and genes with ≥ 2-fold

decrease and minimal oscillation over time (no stage-to-stage increases of >1.5-fold) were retained for analysis. A set of stable maternal reference genes was curated that had <1.25-fold difference in the poly(A)+ samples from 4, 5, 6, and 7 h.p.f. compared to egg and were not significantly changed in Triptolide treatment compared to DMSO (N = 281 genes).

To calculate codon adaptation index, the method of [125] was used based on codon usage frequencies of the top 100 expressed maternally contributed mRNA that were not cleared. For motif analysis, 3'UTRs were extracted from Hsym v1.0 gene annotations, discarding UTRs ≤ 20 nts. The MEME motif finder [85] was run in discriminative mode using the 3'UTRs of early or late cleared genes as foreground and the 3'UTRs of stable genes as background with parameters -rna -nmotifs 20 -mod anr -minw 6 -maxw 12.

To predict miRNAs, a set of reference cnidarian miRNAs was curated as follows: Nematostella and Hydra mature miRNA sequences from miRBase release 22.1 [87], the Hydractinia miR-2022 sequence from [126], and *H. echinata* miRNAs downloaded from the Hydractinia Genome Project [90]. These were used for blastn search against the novel Stringtie genes with parameters -evalue 0.1 -ungapped -word_size 7. Significant hits were overlapped with RNA secondary structure stem-loop regions as predicted by Vienna RNALfold -z [88], and refined structures were predicted for promising candidates with flanking sequence using Vienna RNAfold. Predicted targets were identified by performing blastn on the candidate mature miRNA sequences against cleared and stable mRNA with parameters -evalue 10 -ungapped -strand minus -word_size 4, and manually verified to take potential G:U pairing into account.

## Supporting information

**S1 Fig. rRNA-depletion RNA-seq facilitates detection of genome activation. (A)** Biplot comparing total RNA-seq (no selection) to poly(A)+ selected RNA-seq. **(B)** Proportion of sequencing reads mapping to rRNA without selection, with poly(A)+ selection, and with rRNA depletion at 1 hour post fertilization (h.p.f.). **(C)** Browser tracks over the composite 45S rRNA locus showing RNA-seq coverage in the different selection strategies. **(D)** Browser track showing a predicted array of 45S genes on the genome scaffold HyS0316. **(E)** Heatmaps showing intronic RNA-seq coverage of activated genes over time. RPM = reads per million.
(TIF)

**S2 Fig. *H. symbiolongicarpus* may encode a novel histone H1. (A)** Browser tracks showing two identical novel H1 genes (H1.4) and RNA-seq coverage over time. **(B)** Multiple alignment of the amino acid sequences of human H1.1 (top), *Hydra vulgaris* H1A-like (middle) and the novel *H. symbiolongicarpus* H1.4 (bottom). The CD-Search annotated linker histone domain is marked in red.
(TIF)

**S3 Fig. H4K20 methyltransferases are variably expanded in non-bilaterians. (A)** Protein domain schematics for predicted H4K20 methyltransferases in other cnidarians, showing positions of the SETD8 SET domain (orange), as predicted by CD-Search. Reciprocal best BLAST hits to Hydractinia H4K20 methyltransferases are annotated to the right; embryonic-expressed genes are bolded. **(B)** Predicted H4K20 methyltransferases in *M. leidyi*.
(TIF)

**S4 Fig. H4K20 methyltransferases are expanded in sea urchin. (A)** Protein domain schematics for predicted H4K20 methyltransferases in sea urchin, showing positions of the SETD8 SET domain (orange), as predicted by CD-Search. **(B)** Heatmap showing expression levels of the H4K20 methyltransferases over sea urchin development, using the data of Khor et al 2021. h.p.f. = hours post fertilization. **(C)** Schematics showing BLAST similarity between sea urchin

(top in each pair) and Hydractinia H4K20 methyltransferases. BLAST E-values for each significant high-scoring pair (pink shaded regions) are annotated. SET domains are orange. Three nearly identical sea urchin genes (lower right) have the same degree of BLAST similarity to HyS0109.11, so only one representative schematic is shown.
(TIF)

**S5 Fig. Unannotated expressed genes are mostly non-coding. (A)** Comparison of activation levels of 10-kb windows tiled across the genome at 3 hours post fertilization (h.p.f.) versus egg (x axis) and 4 h.p.f. versus egg (y axis). Windows with major levels of activation are accounted for by histone genes (red triangles), except for two windows that have predicted ribosomal RNA genes. **(B)** Biplots of summed expression across the predicted noncoding gene classes uncovered by Stringtie transcriptome assembly.
(TIF)

**S6 Fig. Stage-specific transcription factors are expressed after genome activation.** Heatmap showing expression patterns of non-maternal stage-specific transcription factors. Gene names are the best BLAST hit to UniProtKB / SwissProt, DNA binding domains are in parentheses.
(TIF)

**S7 Fig. Evaluation of different potential maternal clearance mechanisms. (A)** Table of maternal expression levels of predicted deadenylase factors as identified by BLAST search. **(B)** Biplot showing expression levels in untreated versus DMSO vehicle embryos. Orange points are genes with significant activation over time in wild-type embryos. **(C)** Biplot showing failed activation of wild-type activated genes (orange) with Triptolide treatment. Two of these genes have prominently higher expression upon Triptolide treatment, HyS0422.6 and HyS4764.1. These are both single-exon genes similar to retrovirus-related reverse transcriptase genes, suggesting that these might actually be transposon sequences and not transcribed by Pol II. Conversely, these transposons may be inhibited by mechanisms requiring genome activation, which would account for their up-expression upon Triptolide treatment. TPM = transcripts per million. **(D)** Comparison of codon adaptation index (CAI) between cleared mRNA and a set of stable mRNA, showing no significant decreased CAI associated with clearance. **(E, F)** Top hits from the CISBP-RNA database for each of the MEME motifs enriched in cleared genes.
(TIF)

**S8 Fig. Potential miRNA-mediated regulation of maternal clearance. (A)** Maternal expression levels for predicted components of the miRNA biogenesis pathway and the RNA-induced silencing complex. **(B)** Alignment of predicted *H. symbiolongicarpus* mature miRNAs with *H. echinata* miRNAs. **(C)** A second transcript that contains the predicted miR-B miRNA sequences, but in a structural context inconsistent with canonical Dicer processing (canonical Dicer cleavage sites marked by red arrows). Asterisk marks a base difference compared to *H. echinata*. **(D)** The full duplex structure in which predicted miRNA miR-C is found, suggesting it is not a Drosha substrate. **(E, F)** Predicted mRNA targets, as recovered by BLAST sequence similarity, showing potential base pairing configuration with the mature miRNAs. **(G)** RNA-seq expression levels of predicted mRNA targets of miRNAs in the presence or absence of Triptolide, as compared to maternal levels in the egg. None of the differences are significant, by DESeq2 with FDR adjustment.
(TIF)

**S1 Table. Additional figure data.**
(XLSX)

**S2 Table. Oligos for rRNA-depletion.**
(XLSX)

**S3 Table. RNA-seq expression levels.**
(XLSX)

**S4 Table. Activated genes.**
(XLSX)

**S5 Table. All differential expression stats.**
(XLSX)

**S6 Table. S. purpuratus RNA-seq.**
(XLSX)

**S7 Table. Other species RNA-seq.**
(XLSX)

**S8 Table. H4K20me transferase domains.**
(XLSX)

**S9 Table. Stringtie-predicted genes.**
(XLSX)

**S10 Table. Stage-specific genes.**
(XLSX)

**S11 Table. Maternal clearance categories.**
(XLSX)

## Acknowledgments

We thank U. Frank, Febrimarsa, and C. Schnitzler for discussions and advice, P. Rangan and T. Levin for feedback on the manuscript, and S. Sanders for assistance with animal collection. This project used the University of Pittsburgh Health Sciences Core at UPMC Children's Hospital Pittsburgh for sequencing, and was supported by the University of Pittsburgh Center for Research Computing for computational resources.

## Author Contributions

**Conceptualization:** Miler T. Lee.

**Data curation:** Taylor N. Ayers, Miler T. Lee.

**Formal analysis:** Taylor N. Ayers, Miler T. Lee.

**Funding acquisition:** Miler T. Lee.

**Investigation:** Taylor N. Ayers, Matthew L. Nicotra, Miler T. Lee.

**Methodology:** Miler T. Lee.

**Project administration:** Miler T. Lee.

**Resources:** Matthew L. Nicotra.

**Supervision:** Miler T. Lee.

**Validation:** Miler T. Lee.

**Visualization:** Miler T. Lee.

**Writing – original draft:** Miler T. Lee.

**Writing – review & editing:** Taylor N. Ayers, Matthew L. Nicotra, Miler T. Lee.

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
