## [Decision Letter · Decision Letter 0]

12 Jun 2023

Dear Miler,

Thank you very much for submitting your Research Article entitled 'Parallels and contrasts between the cnidarian and bilaterian maternal-to-zygotic transition are revealed in Hydractinia embryos' to PLOS Genetics.

The manuscript was fully evaluated at the editorial level and by independent peer reviewers. The reviewers appreciated the depth of the maternal to zygotic transcriptome study in Hydractinia and its interesting differences and commonalities to bilaterians but identified some concerns that we ask you to address in a revised manuscript.  We therefore ask you to modify the manuscript according to the review recommendations. Your revisions should address the specific points made by each reviewer.

An accompanying reviewer 2 attachment should be included with this email; please notify the journal office if any appear to be missing. They will also be available for download from the link below. You can use this link to log into the system when you are ready to submit a revised version, having first consulted our Submission Checklist.

Yours sincerely,

Mary

Mary C. Mullins

Academic Editor

PLOS Genetics

Gregory P. Copenhaver

Editor-in-Chief

PLOS Genetics

Reviewer's Responses to Questions

**Comments to the Authors:**

Reviewer #1: The manuscript by Ayers et al is an important contribution, significantly expanding our understanding of early embryonic mechanisms to an understudied taxon. Mechanisms that control the maternal to zygotic transition in animals are varied and likely have evolved quickly. Therefore, studies in cnidarians, which occupy a critical phylogenetic position, are important for understanding the common ancient mechanisms from which extant mechanisms evolved. While a handful of studies have addressed this issue superficially in cnidarians, this is the most comprehensive study to date. The study is well conceived, carefully performed and analyzed, and the manuscript is extremely well written.

I have only a small number of comments, all of which are minor:

Figure 1: I would find it helpful to have a diagram of Hydractinia development that correlates the hours post fertilization, the cell number, and the embryonic stage. (Line 93: Is the blastula really 16 cells?)

Figure 5A: I completely understand the point being made in figures 5A-C, but I’m really struggling to understand the specific information being conveyed in the Figure 5A bubble plot. I don’t understand what each square represents in terms of comparing the two data sets. It is also unclear what the colors mean. I think this is the one figure panel in the whole paper that wasn’t immediately clear and could use some updating.

I was very curious about the H4K20 methyltransferase findings! Are H4K20 methyltransferases expanded in other hydroids, or more broadly in the cnidarian phylum? It would be helpful to better understand how these data into the context of the animal tree. Even though the use of pharmacological inhibitors comes with a lot of caveats, it could be interesting to see how the SETD8 inhibitor used in the referenced Hydra study affects MZT in Hydractinia.

Reviewer #2: see attached

Reviewer #3: This manuscript describes a study of embryonic development in the model cnidarian Hydractinia symbiolongicarpus with a focus on activation of the zygotic genome and the dynamics of the maternal mRNA pool present in the egg. This study is of considerable significance because it represents the first thorough and carefully designed investigation of these phenomena in a non-bilaterian organism. As the sister taxon to bilaterians, Cnidaria is a particularly important phylum for carrying out studies of embryonic development. Comparisons of developmental processes in Cnidaria and Bilateria allow us to define how these processes were established at a particularly important stage in the evolution of metazoans.

The authors show that the zygotic genome is activated at the 64 cell stage. They also show that Hydractinia embryos carry out the same maternal mRNA readenylation, deadenylation, and clearance as is seen in the embryos of model bilaterians such as the frog Xenopus laevis. Interestingly, the activation of the zygotic genome is dominated by activation histone gene expression.

The data in this paper are extensive, are of high quality, and are carefully analyzed. Because of the ability to generate transgenic lines, and to edit genes in Hydractinia with CRISPR/Cas, this study has the additional important feature of setting the stage for future detailed functional studies

I have only one concern about the experimental approaches used in this study. The authors make use of Triptolide, which they describe as an inhibitor of RNA Polymerase II. My quick search of the literature indicates that Triptolide has a number of targets in addition to RNA Polymerase II. Does this create difficulties in interpretation of the data generated from the experiments using Triptolide? I would like to see this issue discussed in the paper.

**Have all data underlying the figures and results presented in the manuscript been provided?**

Reviewer #1: Yes

Reviewer #2: Yes

Reviewer #3: Yes

PLOS authors have the option to publish the peer review history of their article (what does this mean?). If published, this will include your full peer review and any attached files.

Reviewer #1: No

Reviewer #2: No

Reviewer #3: **Yes: **Robert E. Steele

---

## [Editor Report · Decision Letter 1]

26 Jun 2023

Dear Miler

We are pleased to inform you that your manuscript entitled "Parallels and contrasts between the cnidarian and bilaterian maternal-to-zygotic transition are revealed in Hydractinia embryos" has been editorially accepted for publication in PLOS Genetics. Congratulations!

Yours sincerely,

Mary

Mary C. Mullins

Academic Editor

PLOS Genetics

Gregory P. Copenhaver

Editor-in-Chief

PLOS Genetics

Comments from the reviewers (if applicable):

**Data Deposition**

http://datadryad.org/submit?journalID=pgenetics&manu=PGENETICS-D-23-00533R1

**Press Queries**

---

## [Editor Report · Acceptance letter]

8 Jul 2023

PGENETICS-D-23-00533R1 

Parallels and contrasts between the cnidarian and bilaterian maternal-to-zygotic transition are revealed in *Hydractinia* embryos 

Dear Dr Lee, 

We are pleased to inform you that your manuscript entitled "Parallels and contrasts between the cnidarian and bilaterian maternal-to-zygotic transition are revealed in *Hydractinia* embryos" has been formally accepted for publication in PLOS Genetics! Your manuscript is now with our production department and you will be notified of the publication date in due course.

With kind regards,

Zsofia Freund

PLOS Genetics

On behalf of:
